# Mixture weights optimisation for Alpha-Divergence Variational Inference

**Kamélia Daudel**[1,2*], **Randal Douc**[3]
1: LTCI, Télécom Paris, Institut Polytechnique de Paris, France
2: Department of Statistics, University of Oxford, United Kingdom
3: SAMOVAR, Télécom SudParis, Institut Polytechnique de Paris, France

## Abstract

This paper focuses on $\alpha$-divergence minimisation methods for Variational Inference. We consider the case where the posterior density is approximated by a mixture model and we investigate algorithms optimising the mixture weights of this mixture model by $\alpha$-divergence minimisation, without any information on the underlying distribution of its mixture components parameters. The Power Descent, defined for all $\alpha \neq 1$, is one such algorithm and we establish in our work the full proof of its convergence towards the optimal mixture weights when $\alpha < 1$. Since the $\alpha$-divergence recovers the widely-used exclusive Kullback-Leibler when $\alpha \to 1$, we then extend the Power Descent to the case $\alpha = 1$ and show that we obtain an Entropic Mirror Descent. This leads us to investigate the link between Power Descent and Entropic Mirror Descent: first-order approximations allow us to introduce the Rényi Descent, a novel algorithm for which we prove an $O(1/N)$ convergence rate. Lastly, we compare numerically the behavior of the unbiased Power Descent and of the biased Rényi Descent and we discuss the potential advantages of one algorithm over the other.

## 1 Introduction

Bayesian Inference involves being able to compute or sample from the posterior density. For many useful models, the posterior density can only be evaluated up to a normalisation constant and we must resort to approximation methods.

One major category of approximation methods is Variational Inference, a wide class of optimisation methods which introduce a simpler *variational* family $\mathcal{Q}$ and use it to approximate the posterior density (see for example Variational Bayes [1, 2] and Stochastic Variational Inference [3]). The crux of these methods consists in finding the best approximation of the posterior density among the family $\mathcal{Q}$ in the sense of a certain measure of dissimilarity, most typically the *exclusive* Kullback-Leibler divergence.

However, the exclusive Kullback-Leibler divergence is known to have some undesirable properties, e.g. posterior variance underestimation, difficulty to capture multimodal posterior densities [4, 5, 6]. As a consequence, the $\alpha$-divergence [7, 8] and Rényi's $\alpha$-divergence [9, 10] have gained a lot of attention recently and are presented as more general alternatives that permit to bypass the issues of the exclusive Kullback-Leibler when $\alpha < 1$ [11, 12, 13, 14, 15, 16, 17, 18, 19, 20, 21, 22].

Noticeably, [19] introduced the $(\alpha, \Gamma)$-descent, a general family of gradient-based algorithms that approximate the posterior density by a mixture model and that are able to optimise the *mixture weights* of this mixture model by $\alpha$-divergence minimisation. The benefit of these types of algorithms is that (i) they expand the traditional parametric variational family to better capture complex (e.g.

---

*Corresponding author: kamelia.daudel@stats.ox.ac.uk

35th Conference on Neural Information Processing Systems (NeurIPS 2021).

multimodal) posterior densities (ii) they allow, in an Sequential Monte Carlo fashion [23], to select the mixture components according to their overall importance in the set of components parameters, without needing to know the distribution of the mixture components parameters. The $(\alpha, \Gamma)$-descent step is then paired up with an Exploration step that acts on the mixture components parameters, so that a complete algorithm is obtained by alternating between these two steps [19].

The key algorithm of [19] is the Power Descent, which sets $\Gamma(v) = [(\alpha - 1)v + 1]^{\eta/(1-\alpha)}$ with $\alpha \neq 1$ and $\eta > 0$ in the $(\alpha, \Gamma)$-descent. Indeed, when $\alpha < 1$ and as the dimension increases, numerical experiments in [19] show that the Power Descent outperforms the Entropic Mirror Descent (a classical algorithm from the optimisation litterature corresponding to $\Gamma(v) = e^{-\eta v}$ with $\eta > 0$).

Nonetheless, the global convergence of the Power Descent algorithm when $\alpha < 1$, as stated in [19, Theorem 4], assumes the existence of the limit and does not provide conditions that satisfy this assumption. Furthermore, even though the convergence towards the global optimum is derived, there is no convergence rate available for the Power Descent when $\alpha < 1$. After recalling the basics of the Power Descent algorithm in Section 2, we make the following contributions in this paper:

• In Section 3, we prove the full convergence of the Power Descent algorithm towards the optimal mixture weights when $\alpha < 1$ (Theorem 2).

• Since the $\alpha$-divergence becomes the traditional exclusive Kullback-Leibler when $\alpha \to 1$, we next investigate the extension of the Power Descent to the case $\alpha = 1$ in Proposition 1 from Section 4 and we obtain that the Power Descent recovers an Entropic Mirror Descent performing exclusive Kullback-Leibler minimisation.

• We further study the connections between Power Descent and Entropic Mirror Descent by considering first-order approximations, in the hope of finding an algorithm close to the Power Descent and for which we can prove a convergence rate when $\alpha < 1$. As a result, we are able to go beyond the $(\alpha, \Gamma)$-descent framework from [19] by introducing an algorithm closely-related to the Power Descent, that is proved to converge at an $O(1/N)$ rate when $\alpha < 1$ as a consequence of Theorem 3 from Section 4. We call this algorithm the *Rényi Descent* due to the link we establish between the Rényi Descent and Entropic Mirror Descent steps applied to the Variational Rényi bound [12].

• Finally, we run some numerical experiments in Section 5 to compare the behavior of the Power Descent and of the Rényi Descent altogether, before discussing the potential benefits of one approach over the other.

## 2 Background

This section reviews the Power Descent algorithm from [19] and details how it is applied to mixture weights optimisation in a Variational Inference context.

**Notation and problem statement.** Let $(Y, \mathcal{Y}, \nu)$ be a measured space, where $\nu$ is a $\sigma$-finite measure on $(Y, \mathcal{Y})$. Assume that we are given some observed data $\mathscr{D}$ generated from a probabilistic model $p(\mathscr{D}|y)$ that is parameterised by a latent variable $y \in Y$. Further assuming that $y$ is drawn from a certain prior $p_0(y)$, the posterior density of the latent variable $y$ given the data $\mathscr{D}$ is then defined by:

$$p(y|\mathscr{D}) = \frac{p(y, \mathscr{D})}{p(\mathscr{D})} = \frac{p_0(y)p(\mathscr{D}|y)}{p(\mathscr{D})} \, ,$$

where the normalisation constant $p(\mathscr{D}) = \int_Y p_0(y)p(\mathscr{D}|y)\nu(\mathrm{d}y)$ is called the *marginal likelihood* or *model evidence*. For many complex models arising in Bayesian Inference, we do not know how to directly sample from the posterior density, nor do we know the value of the marginal likelihood.

To remedy this problem, the Power Descent is a Variational Inference algorithm that introduces a certain measurable space $(T, \mathcal{T})$ and offers to approximate the posterior density by a variational family $\mathcal{Q}$ of the form

$$\mathcal{Q} = \left\{ y \mapsto \int_T \mu(\mathrm{d}\theta)k(\theta, y) \; : \; \mu \in \mathsf{M} \right\} \, , \tag{1}$$

where $\mathsf{M}$ is a convenient subset of $\mathrm{M}_1(T)$, the set of probability measures on $(T, \mathcal{T})$. In doing so, it extends the usual parametric variational family

$$\mathcal{Q} = \{ y \mapsto k(\theta, y) \; : \; \theta \in T \}$$

since it amounts to putting a prior over the parameter $\theta$ in the form of a probability measure. In particular, one strong motivation behind the variational family (1) is that it recovers mixture models as a special case. To see this, let $J \in \mathbb{N}^\star$, $\Theta = (\theta_1, \ldots, \theta_J) \in \mathsf{T}^J$ and define the simplex of $\mathbb{R}^J$ by:

$$\mathcal{S}_J = \left\{ \boldsymbol{\lambda} = (\lambda_1, \ldots, \lambda_J) \in \mathbb{R}^J \ : \ \forall j \in \{1, \ldots, J\}, \ \lambda_j \geqslant 0 \text{ and } \sum_{j=1}^{J} \lambda_j = 1 \right\} . \tag{2}$$

Observe then that when $\mu$ is chosen as a weighted sum of Dirac measures, i.e. $\mu = \sum_{j=1}^{J} \lambda_j \delta_{\theta_j}$ with $\boldsymbol{\lambda} \in \mathcal{S}_J$, the variational family (1) becomes

$$\mathcal{Q} = \left\{ y \mapsto \sum_{j=1}^{J} \lambda_j k(\theta_j, y) \ : \ \boldsymbol{\lambda} \in \mathcal{S}_J \right\} , \tag{3}$$

which corresponds to the class of mixture models parameterised by the mixture weights $\boldsymbol{\lambda}$ and with fixed mixture components parameters $\Theta$. Here, the $j$-th component $y \mapsto k(\theta_j, y)$ could for example be a multivariate Gaussian density with mean $\theta_j$ and known covariance matrix, but one strength of the Power Descent is that it is not limited to a specific choice for $k$. We will refer to $k$ as a *kernel density* thereafter and we now introduce some more notation, before stating the optimisation problem solved by the Power Descent.

Denote by $\mathbb{P}$ the probability measure on $(\mathsf{Y}, \mathcal{Y})$ with corresponding density $p(\cdot | \mathscr{D})$ with respect to $\nu$ and by $K : (\theta, A) \mapsto \int_A k(\theta, y) \nu(\mathrm{d}y)$ the Markov transition kernel on $\mathsf{T} \times \mathcal{Y}$ with kernel density $k$ defined on $\mathsf{T} \times \mathsf{Y}$. For all $\mu \in \mathrm{M}_1(\mathsf{T})$ and all $y \in \mathsf{Y}$, we denote $\mu k(y) = \int_\mathsf{T} \mu(\mathrm{d}\theta) k(\theta, y)$ and we let $\mu K$ denote the probability measure with density $\mu k$ with respect to $\nu$. Then, letting $\alpha \in \mathbb{R}$ and assuming that $\mu K$ and $\mathbb{P}$ are both absolutely continuous with respect to $\nu$, the $\alpha$-divergence between $\mu K$ and $\mathbb{P}$ (extended by continuity to the cases $\alpha = 0$ and $\alpha = 1$ as in [24]) is given by

$$D_\alpha(\mu K || \mathbb{P}) = \int_\mathsf{Y} f_\alpha \left( \frac{\mu k(y)}{p(y | \mathscr{D})} \right) p(y | \mathscr{D}) \nu(\mathrm{d}y) ,$$

where $f_\alpha$ is the convex function on $(0, +\infty)$ defined by $f_0(u) = u - 1 - \log(u)$, $f_1(u) = 1 - u + u \log(u)$ and $f_\alpha(u) = \frac{1}{\alpha(\alpha-1)} [u^\alpha - 1 - \alpha(u-1)]$ for all $\alpha \in \mathbb{R} \setminus \{0, 1\}$.

Letting $\alpha \neq 1$, the Power Descent seeks to solve the Variational Inference optimisation problem

$$\inf_{\mu \in \mathsf{M}} D_\alpha(\mu K || \mathbb{P}) , \tag{4}$$

meaning that for $\mu = \sum_{j=1}^{J} \lambda_j \delta_{\theta_j}$ (with corresponding approximating family (3)), it seeks to optimise $\boldsymbol{\lambda}$ by $\alpha$-divergence minimisation i.e. to select mixture components according to their overall importance in the set of components parameters. More generally, the Power Descent aims at solving

$$\inf_{\mu \in \mathsf{M}} \Psi_\alpha(\mu; p) , \tag{5}$$

where $p$ is any measurable positive function on $(\mathsf{Y}, \mathcal{Y})$ and where for all $\mu \in \mathrm{M}_1(\mathsf{T})$, $\Psi_\alpha(\mu; p) = \int_\mathsf{Y} f_\alpha (\mu k(y)/p(y)) p(y) \nu(\mathrm{d}y)$. Crucially, (4) can be reframed as an instance of (5) that sets $p = p(\cdot, \mathscr{D})$ in (5) and that thus does not involve the unknown normalising constant $p(\mathscr{D})$ anymore (see Appendix A.1). In the following, the dependency on $p$ in $\Psi_\alpha$ may be dropped throughout the paper for notational ease when no ambiguity occurs and we next present the Power Descent algorithm.

**The Power Descent algorithm.** Let $\alpha \neq 1$. Given an initial probability measure $\mu_1 \in \mathrm{M}_1(\mathsf{T})$ such that $\Psi_\alpha(\mu_1) < \infty$, $\eta > 0$ and $\kappa$ such that $(\alpha - 1)\kappa \geqslant 0$, the Power Descent algorithm introduced in [19] is an iterative scheme which builds the sequence of probability measures $(\mu_n)_{n \in \mathbb{N}^\star}$

$$\mu_{n+1} = \mathcal{I}_\alpha(\mu_n) , \qquad n \in \mathbb{N}^\star , \tag{6}$$

where for all $\mu \in \mathrm{M}_1(\mathsf{T})$, the one-step transition $\mu \mapsto \mathcal{I}_\alpha(\mu)$ is given by Algorithm 1 and where for all $v \in \mathrm{Dom}_\alpha$, $\Gamma(v) = [(\alpha - 1)v + 1]^{\eta/(1-\alpha)}$ [and $\mathrm{Dom}_\alpha$ denotes an interval of $\mathbb{R}$ such that for all $\theta \in \mathsf{T}$ and all $\mu \in \mathrm{M}_1(\mathsf{T})$, $b_{\mu,\alpha}(\theta) + \kappa$ and $\mu(b_{\mu,\alpha}) + \kappa \in \mathrm{Dom}_\alpha$].

**Algorithm 1:** *Power descent one-step transition* $(\Gamma(v) = [(\alpha - 1)v + 1]^{\eta/(1-\alpha)})$

---

1. Expectation step : $b_{\mu,\alpha}(\theta) = \int_{\mathsf{Y}} k(\theta,y) f'_\alpha \left( \dfrac{\mu k(y)}{p(y)} \right) \nu(\mathrm{d}y)$

2. Iteration step : $\mathcal{I}_\alpha(\mu)(\mathrm{d}\theta) = \dfrac{\mu(\mathrm{d}\theta) \cdot \Gamma(b_{\mu,\alpha}(\theta) + \kappa)}{\mu(\Gamma(b_{\mu,\alpha} + \kappa))}$

---

The Power Descent is known to have a *gradient-based* structure since $\theta \mapsto b_{\mu,\alpha}(\theta)$ acts as the gradient of $\mu \mapsto \Psi_\alpha(\mu)$ and $\eta$ plays the role of a learning rate [19]. Furthermore, a remarkable property of the Power Descent is that under (A1) as defined below

(A1) The density kernel $k$ on $\mathsf{T} \times \mathsf{Y}$, the function $p$ on $\mathsf{Y}$ and the $\sigma$-finite measure $\nu$ on $(\mathsf{Y}, \mathcal{Y})$ satisfy, for all $(\theta, y) \in \mathsf{T} \times \mathsf{Y}$, $k(\theta, y) > 0$, $p(y) > 0$ and $\int_{\mathsf{Y}} p(y)\nu(\mathrm{d}y) < \infty$.

the Power Descent ensures a monotonic decrease in $\Psi_\alpha$ at each step for all $\eta \in (0, 1]$ (this result is a special case of [19, Theorem 1] with $\Gamma(v) = [(\alpha - 1)v + 1]^{\eta/(1-\alpha)}$ that is recalled in Theorem 4 of Appendix A.2 for the sake of completeness). Under the additional assumptions that $\kappa > 0$ and

$$\sup_{\theta \in \mathsf{T}, \mu \in \mathrm{M}_1(\mathsf{T})} |b_{\mu,\alpha}| < \infty \quad \text{and} \quad \Psi_\alpha(\mu_1) < \infty \,, \tag{7}$$

the Power Descent is also known to converge towards its optimal value at an $O(1/N)$ rate when $\alpha > 1$ [19, Theorem 3]. On the other hand, when $\alpha < 1$, the convergence towards the optimum as written in [19] holds under different assumptions including

(A2)  (i) $\mathsf{T}$ is a compact metric space and $\mathcal{T}$ is the associated Borel $\sigma$-field;
  (ii) for all $y \in \mathsf{Y}$, $\theta \mapsto k(\theta, y)$ is continuous;
  (iii) we have $\int_{\mathsf{Y}} \sup_{\theta \in \mathsf{T}} k(\theta, y) \times \sup_{\theta' \in \mathsf{T}} \left( \frac{k(\theta', y)}{p(y)} \right)^{\alpha-1} \nu(\mathrm{d}y) < \infty$.

  If $\alpha = 0$, assume in addition that $\int_{\mathsf{Y}} \sup_{\theta \in \mathsf{T}} \left| \log \left( \frac{k(\theta, y)}{p(y)} \right) \right| p(y)\nu(\mathrm{d}y) < \infty$.

so that [19, Theorem 4], that is recalled below under the form of Theorem 1, states the convergence of the Power Descent algorithm towards the global optimum.

**Theorem 1** ([19, Theorem 4]). *Assume* (A1) *and* (A2). *Let* $\alpha < 1$ *and let* $\kappa \leqslant 0$. *Then, for all* $\mu \in \mathrm{M}_1(\mathsf{T})$, $\Psi_\alpha(\mu) < \infty$ *and any* $\eta > 0$ *satisfies* $0 < \mu(\Gamma(b_{\mu,\alpha} + \kappa)) < \infty$. *Further assume that* $\eta \in (0, 1]$ *and that there exist* $\mu_1, \mu^\star \in \mathrm{M}_1(\mathsf{T})$ *such that the (well-defined) sequence* $(\mu_n)_{n \in \mathbb{N}^\star}$ *defined by* (6) *weakly converges to* $\mu^\star$ *as* $n \to \infty$. *Finally, denote by* $\mathrm{M}_{1,\mu_1}(\mathsf{T})$ *the set of probability measures dominated by* $\mu_1$. *Then the following assertions hold*

  *(i)* $(\Psi_\alpha(\mu_n))_{n \in \mathbb{N}^\star}$ *is nonincreasing,*

  *(ii)* $\mu^\star$ *is a fixed point of* $\mathcal{I}_\alpha$,

  *(iii)* $\Psi_\alpha(\mu^\star) = \inf_{\zeta \in \mathrm{M}_{1,\mu_1}(\mathsf{T})} \Psi_\alpha(\zeta)$.

The above result assumes there must exist $\mu_1, \mu^\star \in \mathrm{M}_1(\mathsf{T})$ such that the sequence $(\mu_n)_{n \in \mathbb{N}^\star}$ defined by (6) weakly converges to $\mu^\star$ as $n \to \infty$, that is it assumes the limit already exists. Our first contribution consists in showing that this assumption can be alleviated when $\mu$ is chosen a weighted sum of Dirac measures, that is when we seek to perform mixture weights optimisation by $\alpha$-divergence minimisation. In doing so, we provide additionnal theoretical justification behind the use of the Power Descent when $\alpha < 1$, which is crucial in practice for reasons already outlined in the introduction and further detailed for the particular case of the Power Descent in Appendix A.3.

## 3 Convergence of the Power Descent algorithm in the mixture case

Before we state our convergence result, let us first make two comments on the assumptions from Theorem 1 that shall be retained in our upcoming convergence result.

A first comment is that (A1) is mild since the assumption that $p(y) > 0$ for all $y \in \mathsf{Y}$ can be discarded and is kept for convenience [19, Remark 4]. A second comment is that (A2) is also mild and covers (7) as it amounts to assuming that $\theta \mapsto b_{\mu,\alpha}(\theta)$ and $\mu \mapsto \Psi_\alpha(\mu)$ are uniformly bounded with respect to $\mu$ and $\theta$. To see this, we give below an example for which (A2) is satisfied.

**Example 1.** *Consider the case* $\mathsf{Y} = \mathbb{R}^d$ *with* $\alpha \in [0,1)$. *Let* $r > 0$ *and let* $\mathsf{T} = \mathcal{B}(0,r) \subset \mathbb{R}^d$. *Furtheremore, let* $K_h$ *be a Gaussian transition kernel with bandwidth* $h$ *and denote by* $k_h$ *its associated kernel density. Finally, let* $p$ *be a mixture density of two d-dimensional Gaussian distributions multiplied by a positive constant* $c$ *such that* $p(y) = c \times [0.5\mathcal{N}(y; \theta_1^\star, \boldsymbol{I_d}) + 0.5\mathcal{N}(y; \theta_2^\star, \boldsymbol{I_d})]$ *for all* $y \in \mathsf{Y}$, *where* $\theta_1^\star, \theta_2^\star \in \mathsf{T}$ *and* $\boldsymbol{I_d}$ *is the identity matrix. Then,* (A2) *holds (see Appendix B.1).*

Next, we introduce some notation that are specific to the case of mixture models we aim at studying in this section (and which, as seen previously, corresponds to choosing $\mu$ as a weighted sum of Dirac measures in (1) yielding (3)). For this purpose, let $J \in \mathbb{N}^\star$, let $\Theta = (\theta_1, \ldots, \theta_J) \in \mathsf{T}^J$ be fixed, let $\alpha \neq 1$, $\eta > 0$, $\kappa$ be such that $(\alpha - 1)\kappa \geqslant 0$ and recall that the simplex of $\mathbb{R}^J$ defined in (2) is denoted by $\mathcal{S}_J$. We define $\mathcal{S}_J^+ = \{\boldsymbol{\lambda} \in \mathcal{S}_J : \forall j \in \{1, \ldots, J\}, \lambda_j > 0\}$ and for all $\boldsymbol{\lambda} \in \mathcal{S}_J$ and we also define $\mu_{\boldsymbol{\lambda},\Theta} = \sum_{j=1}^J \lambda_j \delta_{\theta_j}$. Now letting $(\mu_n)_{n \in \mathbb{N}^\star}$ be defined by $\mu_1 = \mu_{\boldsymbol{\lambda},\Theta}$ and (6), an immediate induction yields that for every $n \in \mathbb{N}^\star$, $\mu_n$ can be expressed as $\mu_n = \sum_{j=1}^J \lambda_{j,n} \delta_{\theta_j}$ where $\boldsymbol{\lambda}_n = (\lambda_{1,n}, \ldots, \lambda_{J,n}) \in \mathcal{S}_J$ satisfies the initialisation $\boldsymbol{\lambda}_1 = \boldsymbol{\lambda}$ and the update formula:

$$\boldsymbol{\lambda}_{n+1} = \mathcal{I}_\alpha^{\mathrm{mixt}}(\boldsymbol{\lambda}_n), \ n \in \mathbb{N}^\star, \tag{8}$$

with

$$\mathcal{I}_\alpha^{\mathrm{mixt}}(\boldsymbol{\lambda}) := \left( \frac{\lambda_j \left[(\alpha - 1)(b_{\mu_{\boldsymbol{\lambda},\Theta},\alpha}(\theta_j) + \kappa) + 1\right]^{\frac{\eta}{1-\alpha}}}{\sum_{\ell=1}^J \lambda_\ell \left[(\alpha - 1)(b_{\mu_{\boldsymbol{\lambda},\Theta},\alpha}(\theta_\ell) + \kappa) + 1\right]^{\frac{\eta}{1-\alpha}}} \right)_{1 \leqslant j \leqslant J}, \quad \boldsymbol{\lambda} \in \mathcal{S}_J.$$

Finally, let us rewrite (A2) in the simplified case where $\mu = \mu_{\boldsymbol{\lambda},\Theta}$, which gives (A3) below.

(A3) (i) For all $y \in \mathsf{Y}$, $\theta \mapsto k(\theta, y)$ is continuous;

(ii) we have $\int_{\mathsf{Y}} \max_{1 \leqslant j \leqslant J} k(\theta_j, y) \times \max_{1 \leqslant j' \leqslant J} \left( \frac{k(\theta_{j'}, y)}{p(y)} \right)^{\alpha - 1} \nu(\mathrm{d}y) < \infty$.

If $\alpha = 0$, we assume in addition that $\int_{\mathsf{Y}} \max_{1 \leqslant j \leqslant J} \left| \log \left( \frac{k(\theta_j, y)}{p(y)} \right) \right| p(y) \nu(\mathrm{d}y) < \infty$.

We then have the following theorem, which establishes the full proof of the global convergence towards the optimum for the mixture weights when $\alpha < 1$.

**Theorem 2.** *Assume* (A1) *and* (A3). *Let* $\alpha < 1$, *let* $\Theta = (\theta_1, \ldots, \theta_J) \in \mathsf{T}^J$ *be fixed and let* $\kappa$ *be such that* $\kappa \leqslant 0$. *Then for all* $\boldsymbol{\lambda} \in \mathcal{S}_J$, $\Psi_\alpha(\mu_{\boldsymbol{\lambda},\Theta}) < \infty$ *and for any* $\eta > 0$ *the sequence* $(\boldsymbol{\lambda}_n)_{n \in \mathbb{N}^\star}$ *defined by* $\boldsymbol{\lambda}_1 \in \mathcal{S}_J$ *and* (8) *is well-defined. If in addition* $(\boldsymbol{\lambda}_1, \eta) \in \mathcal{S}_J^+ \times (0, 1]$ *and* $\{K(\theta_1, \cdot), \ldots, K(\theta_J, \cdot)\}$ *are linearly independent, then*

(i) $(\Psi_\alpha(\mu_{\boldsymbol{\lambda}_n,\Theta}))_{n \in \mathbb{N}^\star}$ *is nonincreasing,*

(ii) *the sequence* $(\boldsymbol{\lambda}_n)_{n \in \mathbb{N}^\star}$ *converges to some* $\boldsymbol{\lambda}_\star \in \mathcal{S}_J$ *which is a fixed point of* $\mathcal{I}_\alpha^{\mathrm{mixt}}$,

(iii) $\Psi_\alpha(\mu_{\boldsymbol{\lambda}_\star,\Theta}) = \inf_{\boldsymbol{\lambda}' \in \mathcal{S}_J} \Psi_\alpha(\mu_{\boldsymbol{\lambda}',\Theta})$.

The proof of this result builds on Theorem 1 and Theorem 4 of Appendix A.2 and is deferred to Appendix B.2. Notice that since $\Psi_\alpha$ depends on $\boldsymbol{\lambda}$ through $\mu_{\boldsymbol{\lambda},\Theta}K$, an identifiably condition was to be expected in order to achieve the convergence of the sequence $(\boldsymbol{\lambda}_n)_{n \in \mathbb{N}^\star}$ in Theorem 2. Following Example 1, this identifiably condition notably holds for $J \leqslant d$ under the assumption that the $\theta_1, \ldots, \theta_J$ are full-rank.

We thus have the convergence of the Power Descent under less stringent conditions when $\alpha < 1$ and when we consider the particular case of mixture models. This algorithm can easily become feasible for any choice of kernel $K$ by resorting to an unbiased estimator of $(b_{\mu_{\boldsymbol{\lambda}_n,\Theta},\alpha}(\theta_j))_{1 \leqslant j \leqslant J}$ in the update formula (8) (see Algorithm 3 of Appendix B.3).

Nevertheless, contrary to the case $\alpha > 1$ we still do not have a convergence rate for the Power Descent when $\alpha < 1$. Furthermore, the important case $\alpha \to 1$, which corresponds to performing

exclusive Kullback-Leibler minimisation, is not covered by the Power Descent algorithm. In the next section, we extend the Power Descent to the case $\alpha = 1$. As we shall see, this will lead us to investigate the connections between the Power Descent and the Entropic Mirror Descent beyond the $(\alpha, \Gamma)$-descent framework from [19]. As a result, we will introduce a novel algorithm closely-related to the Power Descent that yields an $O(1/N)$ convergence rate when $\mu = \mu_{\lambda,\Theta}$ and $\alpha < 1$ (and more generally when $\mu \in M_1(\mathsf{T})$ and $\alpha \neq 1$).

## 4 Power Descent and Entropic Mirror Descent

Recall from Section 2 that the Power Descent is defined for all $\alpha \neq 1$. In this section, we first establish in Proposition 1 that the Power Descent can be extended to the case $\alpha = 1$ and that we recover an Entropic Mirror Descent, showing that a deeper connection runs between the two approaches beyond the one identified by the $(\alpha, \Gamma)$-descent framework of [19]. This result relies on typical convergence and differentiability assumptions summarised in (D1) and which are deferred to Appendix C.1, alongside with the proof of Proposition 1.

**Proposition 1** (Limiting case $\alpha \to 1$). *Assume* (A1) *and* (D1). *Let $\eta > 0$ and $\kappa$ be such that $(\alpha - 1)\kappa \geqslant 0$. Then, for all $\mu \in M_1(\mathsf{T})$ and all continuous and bounded real-valued functions $h$ on $\mathsf{T}$, we have that*

$$\lim_{\alpha \to 1} [\mathcal{I}_\alpha(\mu)](h) = [\mathcal{I}_1(\mu)](h) \ ,$$

*where for all $\theta \in \mathsf{T}$, we have set*

$$\mathcal{I}_1(\mu)(\mathrm{d}\theta) = \frac{\mu(\mathrm{d}\theta)e^{-\eta b_{\mu,1}(\theta)}}{\mu\left(e^{-\eta b_{\mu,1}}\right)} \quad and \quad b_{\mu,1}(\theta) = \int_\mathsf{Y} k(\theta, y) \log\left(\frac{\mu k(y)}{p(y)}\right) \nu(\mathrm{d}y) \ , \qquad (9)$$

*and where for all $\zeta \in M_1(\mathsf{T})$, we have used the notation $\zeta h = \int_\mathsf{T} \zeta(\mathrm{d}\theta)h(\theta)$.*

Here, we recognise the one-step transition associated to the Entropic Mirror Descent applied to $\Psi_1$. This algorithm is a special case of [19] with $\Gamma(v) = e^{-\eta v}$ and $\alpha = 1$ and as such, it is known to lead to a systematic decrease in the exclusive Kullback-Leibler divergence and to enjoy an $O(1/N)$ convergence rate under the assumptions that (7) holds and $\eta \in (0, 1)$ [19, Theorem 3].

We have thus obtained that the Power Descent coincides exactly with the Entropic Mirror Descent applied to $\Psi_1$ when $\alpha = 1$ and we now focus on understanding the links between Power Descent and Entropic Mirror Descent when $\alpha \neq 1$. For this purpose, let $\kappa$ be such that $(\alpha - 1)\kappa \geqslant 0$ and let us study first-order approximations of the Power Descent and of the Entropic Mirror Descent applied to $\Psi_\alpha$ when $b_{\mu_n,\alpha}(\theta) \approx \mu_n(b_{\mu_n,\alpha})$ for all $\theta \in \mathsf{T}$.

Letting $\eta > 0$ and following (6), we have that the update formula for the Power Descent is given by

$$\mu_{n+1}(\mathrm{d}\theta) = \frac{\mu_n(\mathrm{d}\theta)\left[(\alpha - 1)(b_{\mu_n,\alpha}(\theta) + \kappa) + 1\right]^{\frac{\eta}{1-\alpha}}}{\mu_n\left(\left[(\alpha - 1)(b_{\mu_n,\alpha} + \kappa) + 1\right]^{\frac{\eta}{1-\alpha}}\right)} \ , \quad n \in \mathbb{N}^\star \ ,$$

and using the first-order approximation $u^{\frac{\eta}{1-\alpha}} \approx v^{\frac{\eta}{1-\alpha}} - \frac{\eta}{\alpha-1}v^{\frac{\eta}{1-\alpha}-1}(u - v)$ with $u = \frac{(\alpha-1)(b_{\mu_n,\alpha}(\theta)+\kappa)+1}{(\alpha-1)(\mu(b_{\mu_n,\alpha})+\kappa)+1}$ and $v = 1$, we can deduce the following approximated update formula

$$\mu_{n+1}(\mathrm{d}\theta) = \mu_n(\mathrm{d}\theta)\left[1 - \frac{\eta}{\alpha - 1}\frac{b_{\mu_n,\alpha}(\theta) - \mu_n(b_{\mu_n,\alpha})}{\mu_n(b_{\mu_n,\alpha}) + \kappa + 1/(\alpha - 1)}\right] \ , \quad n \in \mathbb{N}^\star \ .$$

In addition, the update formula for the Entropic Mirror Descent applied to $\Psi_\alpha$ can be written as

$$\mu_{n+1}(\mathrm{d}\theta) = \frac{\mu_n(\mathrm{d}\theta)\exp\left[-\eta(b_{\mu_n,\alpha}(\theta) + \kappa)\right]}{\mu_n(\exp\left[-\eta(b_{\mu_n,\alpha} + \kappa)\right])} \ , \quad n \in \mathbb{N}^\star \ , \qquad (10)$$

and we obtain in a similar fashion that an approximated version of this iterative scheme is

$$\mu_{n+1}(\mathrm{d}\theta) = \mu_n(\mathrm{d}\theta)\left[1 - \eta\left(b_{\mu_n,\alpha}(\theta) - \mu_n(b_{\mu_n,\alpha})\right)\right] \ , \quad n \in \mathbb{N}^\star \ .$$

As is, the two approximated formulas above do not coincide. This leads us to modify (10) as follows

$$\mu_{n+1}(\mathrm{d}\theta) = \frac{\mu_n(\mathrm{d}\theta)\exp\left[-\eta\frac{b_{\mu_n,\alpha}(\theta)}{(\alpha-1)(\mu_n(b_{\mu_n,\alpha})+\kappa)+1}\right]}{\mu_n\left(\exp\left[-\eta\frac{b_{\mu_n,\alpha}}{(\alpha-1)(\mu_n(b_{\mu_n,\alpha})+\kappa)+1}\right]\right)} \ , \quad n \in \mathbb{N}^\star \ , \qquad (11)$$

that is, it leads us to move from a constant learning rate $\eta$ in (10) to an adaptive learning rate $\eta' = \eta \left[(\alpha - 1)(\mu_n(b_{\mu_n,\alpha}) + \kappa) + 1\right]^{-1}$ in (11). While this change is subtle, a first remark is that [19] only considered a constant learning rate when performing Entropic Mirror Descent steps applied to $\Psi_\alpha$, meaning that we are now deviating from the framework of [19]. A second remark is that we can motivate (11) by observing that it can be seen as an Entropic Mirror Descent too, but this time applied to the objective function defined for all measurable positive function $p$ on $(\mathsf{Y}, \mathcal{Y})$, all $\mu \in \mathrm{M}_1(\mathsf{T})$ and all $\alpha \in \mathbb{R} \setminus \{0, 1\}$ by

$$\Psi_\alpha^{AR}(\mu; p) := \frac{1}{\alpha(\alpha - 1)} \log\left(\int_{\mathsf{Y}} \mu k(y)^\alpha p(y)^{1-\alpha} \nu(\mathrm{d}y) + (\alpha - 1)\kappa\right)$$

(see Appendix C.2 for the derivation of (11) based on the objective function $\Psi_\alpha^{AR}$). Since the function $\Psi_\alpha^{AR}$ can be obtained by applying the increasing transformation

$$u \mapsto \frac{1}{\alpha(\alpha - 1)} \log\left(\alpha(\alpha - 1)u + \alpha + (1 - \alpha)\int_{\mathsf{Y}} p(y)\nu(\mathrm{d}y) + (\alpha - 1)\kappa\right)$$

to the objective function $\Psi_\alpha$, this notably means that our novel Entropic Mirror Descent-based update formula (11) will aim at solving the initial optimisation problem (5) defined in Section 2 (i.e. minimising $\Psi_\alpha(\mu; p)$ with $\mu \in \mathsf{M}$). In addition, when $p = p(\cdot, \mathscr{D})$, $\kappa = 0$ and $\alpha > 0$ in $\Psi_\alpha^{AR}(\mu; p)$, we obtain that minimising $\Psi_\alpha^{AR}(\mu; p)$ w.r.t $\mu$ is equivalent to setting $q = \mu k$ and maximising the Variational Rényi bound $\mathcal{L}_\alpha(q; \mathscr{D})$ from [12] w.r.t $\mu$, since for all probability density $q$,

$$\mathcal{L}_\alpha(q; \mathscr{D}) := \frac{1}{1 - \alpha} \log\left[\mathbb{E}_q\left(\left(\frac{p(y, \mathscr{D})}{q(y)}\right)^{1-\alpha}\right)\right] .$$

As a result, we can classify the scheme (11) as a Rényi's $\alpha$-divergence gradient-based approach [11, 12]. This is in contrast with the framework of [19], that is an $\alpha$-divergence gradient-based approach and for this reason we call the algorithm given by (11) the *Rényi Descent*.

We have thus obtained that the Entropic Mirror Descent applied to $\Psi_\alpha$ does not share the same first-order approximation as the Power Descent, contrary to the newly-introduced Rényi Descent. This might explain why the behavior of the Entropic Mirror Descent applied to $\Psi_\alpha$ and of the Power Descent differed greatly when $\alpha < 1$ in the numerical experiments from [19] (indeed, despite their theoretical connection through the $(\alpha, \Gamma)$-descent framework, the former performs poorly numerically compared to the later as the dimension increases). This also sparks our interest in studying convergence results for the Rényi Descent.

Strikingly, we can prove an $O(1/N)$ convergence rate towards the global optimum for the Rényi Descent. Letting $\kappa' \in \mathbb{R}$, denoting by $\mathrm{Dom}_\alpha^{AR}$ an interval of $\mathbb{R}$ such that for all $\theta \in \mathsf{T}$ and all $\mu \in \mathrm{M}_1(\mathsf{T})$,

$$\frac{b_{\mu,\alpha}(\theta) + 1/(\alpha - 1)}{(\alpha - 1)(\mu(b_{\mu,\alpha}) + \kappa) + 1} + \kappa' \quad \text{and} \quad \frac{\mu(b_{\mu,\alpha}) + 1/(\alpha - 1)}{(\alpha - 1)(\mu(b_{\mu,\alpha}) + \kappa) + 1} + \kappa' \in \mathrm{Dom}_\alpha^{AR}$$

and introducing the assumption on $\eta$

(A4) For all $v \in \mathrm{Dom}_\alpha^{AR}$, $1 - \eta(\alpha - 1)(v - \kappa') \geqslant 0$.

we indeed have the following convergence result.

**Theorem 3.** *Assume* (A1) *and* (A4). *Let $\alpha \neq 1$ and let $\kappa$ be such that $(\alpha - 1)\kappa > 0$. Define $|B|_{\infty,\alpha} := \sup_{\theta \in \mathsf{T}, \mu \in \mathrm{M}_1(\mathsf{T})} |b_{\mu,\alpha}(\theta) + 1/(\alpha - 1)|$ and assume that $|B|_{\infty,\alpha} < \infty$. Moreover, let $\mu_1 \in \mathrm{M}_1(\mathsf{T})$ be such that $\Psi_\alpha(\mu_1) < \infty$. Then, the following assertions hold.*

    *(i) The sequence $(\mu_n)_{n \in \mathbb{N}^\star}$ defined by (11) is well-defined and the sequence $(\Psi_\alpha(\mu_n))_{n \in \mathbb{N}^\star}$ is non-increasing.*

    *(ii) For all $N \in \mathbb{N}^\star$, we have*

$$\Psi_\alpha(\mu_N) - \Psi_\alpha(\mu^\star) \leqslant \frac{L_{\alpha,2}}{N}\left[KL(\mu^\star \| \mu_1) + L\frac{L_{\alpha,3}}{L_{\alpha,1}(\alpha - 1)\kappa}\Delta_1\right] , \tag{12}$$

    *where $\mu^\star$ is such that $\Psi_\alpha(\mu^\star) = \inf_{\zeta \in \mathrm{M}_{1,\mu_1}(\mathsf{T})} \Psi_\alpha(\zeta)$, $\mathrm{M}_{1,\mu_1}(\mathsf{T})$ denotes the set of probability measures dominated by $\mu_1$, $KL(\mu^\star \| \mu_1) = \int_{\mathsf{T}} \log(\mathrm{d}\mu^\star/\mathrm{d}\mu_1)\mathrm{d}\mu^\star$, $\Delta_1 = \Psi_\alpha(\mu_1) - \Psi_\alpha(\mu^\star)$ and $L, L_{\alpha,1}, L_{\alpha,2}, L_{\alpha,3}$ are finite constants defined in (25).*

Table 1: Summary of the theoretical results obtained in this paper compared to [19]

|  | Power Descent | Rényi Descent |
|---|---|---|
| [19] | $\alpha < 1$: convergence under restrictive assumptions; $\alpha > 1$: $O(1/N)$ convergence rate | not covered |
| This paper | $\alpha < 1$: full proof of convergence for mixture weights; extension to $\alpha = 1$ | $O(1/N)$ convergence rate |

The proof of this result is deferred to Appendix C.3 and we present in the next example an application of this theorem to the particular case of mixture models.

**Example 2.** *Let* $\alpha \neq 1$, $\kappa$ *be such that* $(\alpha - 1)\kappa > 0$, $J \in \mathbb{N}^\star$, $\Theta = (\theta_1, \ldots, \theta_J) \in \mathsf{T}^J$, $\mu_1 = J^{-1} \sum_{j=1}^J \delta_{\theta_j}$ *and* $\mathrm{Dom}_\alpha^{AR} = [-\frac{|B|_{\infty,\alpha}}{(\alpha-1)\kappa} + \kappa', \frac{|B|_{\infty,\alpha}}{(\alpha-1)\kappa} + \kappa']$ *with* $\kappa' \in \mathbb{R}$. *In addition, assume that* $1 - \eta|\kappa|^{-1}|B|_{\infty,\alpha} > 0$. *Then, taking* $\kappa' = -3\frac{|B|_{\infty,\alpha}}{(\alpha-1)\kappa}$, *we obtain*

$$\Psi_\alpha(\mu_N) - \Psi_\alpha(\mu^\star) \leqslant \frac{|\alpha - 1|(|B|_{\infty,\alpha} + |\kappa|)}{N} \left[ \frac{\log J}{\eta} + \frac{\sqrt{2\log(J)}|B|_{\infty,\alpha}}{(\alpha-1)\kappa(1 - \eta|\kappa|^{-1}|B|_{\infty,\alpha})} \right] ,$$

*where we have used that* $KL(\mu^\star||\mu_1) \leqslant \log J$, $\Delta_1 \leqslant \sqrt{2\log J}|B|_{\infty,\alpha}$ *and that the constants defined in* (25) *satisfy* $L_{\alpha,2} = \eta^{-1}|\alpha - 1|(|B|_{\infty,\alpha} + |\kappa|)$, $L = \eta^2 e^{\eta \frac{|B|_{\infty,\alpha}}{(\alpha-1)\kappa} - \eta\kappa'}$, $L_{\alpha,3} = e^{\eta \frac{|B|_{\infty,\alpha}}{(\alpha-1)\kappa} + \eta\kappa'}$ *and* $L_{\alpha,1} = (1 - \eta|\kappa|^{-1}|B|_{\infty,\alpha})\eta e^{-\eta \frac{|B|_{\infty,\alpha}}{(\alpha-1)\kappa} - \eta\kappa'}$.

To put things into perspective, as an Entropic Mirror Descent algorithm, the Rényi Descent already enjoys an $O(1/\sqrt{N})$ convergence rate for the sequence $(\Psi_\alpha(N^{-1} \sum_{n=1}^N \mu_n))_{N \in \mathbb{N}^\star}$ under our assumptions and when $\eta$ is proportional to $1/\sqrt{N}$, $N$ being fixed (see [25] or [26, Theorem 4.2.]).

The improvement thus lies in the fact that deriving an $O(1/N)$ convergence rate usually requires stronger assumptions on $\Psi_\alpha$ [26, Theorem 6.2] that we do not need to assume in Theorem 3 thanks to the specific form of $\Psi_\alpha$. Furthermore, due to the monotonicity property, our result only involves the measure $\mu_N$ at time $N$ while typical Entropic Mirror Descent results are expressed in terms of the average $N^{-1} \sum_{n=1}^N \mu_n$.

Finally, observe that the Rényi Descent becomes feasible in practice for any choice of kernel $K$ by letting $\mu$ be a weighted sum of Dirac measures i.e $\mu = \mu_{\boldsymbol{\lambda},\Theta}$ and by resorting to an unbiased estimate of $(b_{\mu,\alpha}(\theta_j))_{1 \leqslant j \leqslant J}$ (see Algorithm 4 of Appendix D.1 for details). The theoretical results we have obtained are summarised in Table 1 and we next move on to numerical experiments.

## 5  Simulation study

The code for all the subsequent numerical experiments is available at `https://github.com/kdaudel/MixtureWeightsAlphaVI`.

Let the target $p$ be a mixture density of two $d$-dimensional Gaussian distributions multiplied by a positive constant $c$ such that $p(y) = c \times [0.5\mathcal{N}(y; -s\boldsymbol{u_d}, \boldsymbol{I_d}) + 0.5\mathcal{N}(y; s\boldsymbol{u_d}, \boldsymbol{I_d})]$, where $\boldsymbol{u_d}$ is the $d$-dimensional vector whose coordinates are all equal to 1, $s = 2$, $c = 2$ and $\boldsymbol{I_d}$ is the identity matrix. Let $K_h$ be a Gaussian transition kernel on $\mathsf{T} \times \mathcal{Y}$ with bandwidth $h$ and let $k_h$ denote its associated kernel density on $\mathsf{T} \times \mathsf{Y}$. Let $J \in \mathbb{N}^\star$ and recall the previously-defined notation $\mu_{\boldsymbol{\lambda},\Theta} = \sum_{j=1}^J \lambda_j \delta_{\theta_j}$ with $\boldsymbol{\lambda} \in \mathcal{S}_J$ and $\Theta \in \mathsf{T}^J$. We consider the variational family described by

$$\mathcal{Q} = \left\{ y \mapsto \mu_{\boldsymbol{\lambda},\Theta} k_h(y) = \sum_{j=1}^J \lambda_j k_h(y - \theta_j) \; : \; \boldsymbol{\lambda} \in \mathcal{S}_J, \Theta \in \mathsf{T}^J \right\} .$$

Following the reasoning of [19], since the Power Descent and the Rényi Descent operate only on the mixture weights $\boldsymbol{\lambda}$ of $\mu_{\boldsymbol{\lambda},\Theta} k_h$ during the optimisation, a fully adaptive algorithm can be obtained by alternating $T$ times between an *Exploitation step* where the mixture weights are optimised and an *Exploration step* where the mixture components parameters are updated, as written in Algorithm 2.

**Algorithm 2:** *Complete Exploitation-Exploration Algorithm*

**Input**: $p$: measurable positive function, $\alpha$: $\alpha$-divergence parameter, $q_0$: initial sampler, $K_h$: Gaussian transition kernel, $T$: total number of iterations, $J$: dimension of the parameter set.
**Output**: Optimised weights $\boldsymbol{\lambda}$ and parameter set $\Theta$.
Draw $\theta_{1,1}, \ldots, \theta_{J,1}$ from $q_0$.
**for** $t = 1 \ldots T$ **do**

> Exploitation step : Set $\Theta = \{\theta_{1,t}, \ldots, \theta_{J,t}\}$. Perform the Power Descent or the Rényi Descent algorithm and obtain the optimised mixture weights $\boldsymbol{\lambda}$.
>
> Exploration step : Perform any Exploration step of our choice and obtain $\theta_{1,t+1}, \ldots, \theta_{J,t+1}$.

Many choices of Exploration steps can be envisioned in Algorithm 2. We first consider the Exploration step used in [19]: $h \propto J^{-1/(4+d)}$ and the particles are updated by i.i.d sampling according to $\mu_{\boldsymbol{\lambda},\Theta}k_h$. Since the Entropic Mirror Descent applied to $\Psi_\alpha$ is known to break down when $d = 16$ for this Exploration step [19], this case is a good starting point in order to investigate if the novel Rényi Descent is a suitable alternative to the Entropic Mirror Descent applied $\Psi_\alpha$ as $d$ increases.

More precisely, at each time $t = 1 \ldots T$, we perform $N$ transitions of either the Power Descent or the Rényi Descent according to Algorithm 3 and 4 of the appendix, in which the initial weights are set to be $[1/J, \ldots, 1/J]$, $\eta = \eta_0/\sqrt{N}$ with $\eta_0 > 0$ and $M$ samples are used in the estimation of $(b_{\mu_{\boldsymbol{\lambda},\Theta},\alpha}(\theta_{j,t}))_{1 \leqslant J}$ at each iteration $n = 1 \ldots N$. We take $J = 100$, $M \in \{100, 1000, 2000\}$, $\alpha = 0.5$, $\kappa = 0$, $\eta_0 = 0.3$ and $q_0$ is a centered normal distribution with covariance matrix $5I_d$. We let $T = 10$, $N = 20$ and we replicate the experiment 100 times independently for each algorithm.

The convergence is assessed using Monte Carlo estimates of the Variational Rényi bound from [12]. The results for the Power Descent and the Rényi Descent are displayed on Figure 1, in which the Entropic Mirror Descent applied to $\Psi_\alpha$ is added as a reference (and additonnal plots in dimension $d < 16$ are provided in Appendix D.2 for the sake of completeness).

Figure 1: Plotted is the average Variational Rényi bound for the Power Descent (PD), the Rényi Descent (RD) and the Entropic Mirror Descent applied to $\Psi_\alpha$ (EMD) in dimension $d = 16$ computed over 100 replicates with $\eta_0 = 0.3$ and $\alpha = 0.5$ and an increasing number of samples $M$.

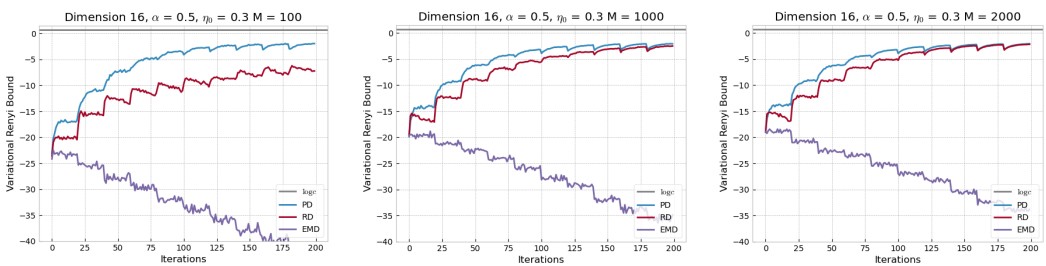

We then observe that the Rényi Descent is indeed better-behaved compared to the Entropic Mirror Descent applied to $\Psi_\alpha$. Furthermore, it matches the performances of the Power Descent as $M$ increases in our numerical experiment, which illustrates the link between the two algorithms we have established in the previous section.

Note that the Exploration step considered in [19] does not optimise over $\Theta$. In an SMC fashion, it indeed resamples in the interesting regions for $\Theta$ (selection) and then applies a Gaussian pertubation (mutation). While we are not trying to discriminate between various choices of Exploration steps in this paper, an important remark is that the choice of the Exploration step in Algorithm 2 will become increasingly important as the dimension increases. For this reason, we have outlined in Appendix D.3 some potentially suitable Exploration steps that optimise over $\Theta$.

In particular, we also ran our numerical experiments with the Exploration step from Appendix D.3.2. The plots for $d = 16$ are available in Appendix D.3.2 and we present below the plots for the higher-dimensional setting where $d = 100$. We observe that the Rényi Descent and the Power Descent do

not break down as we increase the dimension up to $d = 100$ and that they share a similar behavior, which reinforces our conclusions.

Figure 2: Plotted is the average Variational Rényi bound for the Power Descent (PD) and the Rényi Descent (RD) in dimension $d = 100$ computed over 100 replicates with the Exploration Step from Appendix D.3.2, $\eta_0 = 0.3$ and $\alpha = 0.5$ and an increasing number of samples $M$.

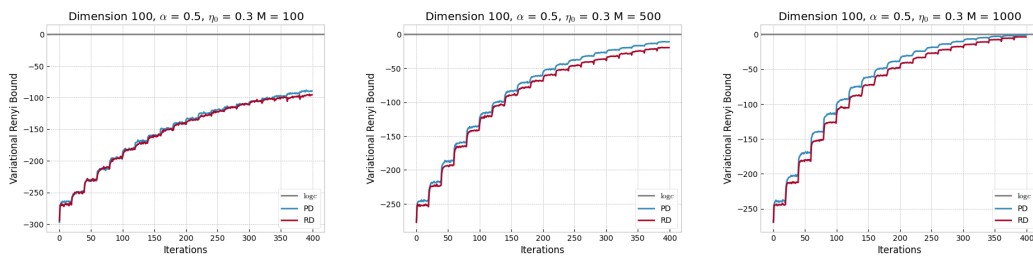

**Discussion**  From a theoretical standpoint, no convergence rate is yet available for the Power Descent algorithm when $\alpha < 1$. An advantage of the novel Rényi Descent algorithm is then that while being close to the Power Descent, it also benefits from the Entropic Mirror Descent optimisation literature so that $O(1/\sqrt{N})$ convergence rates hold, which we have been able to improve to $O(1/N)$ convergence rates.

A practical use of the Power Descent and of the Rényi Descent algorithms requires approximations to handle intractable integrals appearing in the update formulas. As such, the Power Descent applies the function $\Gamma(v) = [(\alpha-1)v+1]^{\eta/(1-\alpha)}$ to an *unbiased* estimator of the translated gradient $b_{\mu,\alpha}(\theta)+\kappa$ before renormalising, while the the Rényi Descent applies the Entropic Mirror Descent function $\Gamma(v) = e^{-\eta v}$ to a *biased* estimator of $b_{\mu_n,\alpha}(\theta)/(\mu_n(b_{\mu_n,\alpha}) + \kappa + 1/(\alpha - 1))$ before renormalising.

Finding which approach is most suitable between biased and unbiased $\alpha$-divergence minimisation is still an open issue in the literature, both theoretically and empirically [17, 18, 21]. Due to the exponentiation, considering the $\alpha$-divergence instead of Rényi's $\alpha$-divergence has for example been said to lead to high-variance gradients [13, 12] and low Signal-to-Noise ratio when $\alpha \neq 0$ [18] during the stochastic gradient descent optimisation.

In that regard, our work sheds light on additional links between unbiased and biased $\alpha$-divergence methods beyond the framework of stochastic gradient descent algorithms, as both the unbiased Power Descent and the biased Rényi Descent share the same first-order approximation.

Our work also contributes towards deriving convergence results of variational objective functions, which is an active area of research [5, 27, 28, 29, 30]. The particularity of our results is then that, in line with the research work started in [19], we focus on mixture weights gradient-based updates in the optimisation procedures, that are carried out via $\alpha$-divergence minimisation and for general choices of kernel $K$. Compared to [19], we also bring into play Rényi's $\alpha$-divergence-based types of updates, which is a novel idea when it comes to mixture weights optimisation by $\alpha$-divergence minimisation.

## 6   Conclusion

We investigated algorithms that can be used to perform mixture weights optimisation for $\alpha$-divergence minimisation regardless of how the mixture components parameters are obtained. We have established the full proof of the convergence of the Power Descent algorithm in the case $\alpha < 1$ when we consider mixture models and bridged the gap with the case $\alpha = 1$. We also introduced a closely-related algorithm called the Rényi Descent. We proved it enjoys an $O(1/N)$ convergence rate and illustrated in practice the proximity between these two algorithms.

Further work could include establishing theoretical results regarding the stochastic version of these two algorithms, as well as providing complementary empirical results comparing the performances of the unbiased $\alpha$-divergence-based Power Descent algorithm to those of the biased Rényi's $\alpha$-divergence-based Rényi Descent. Since our contributions are mainly theoretical, we believe these will not result in any negative societal impacts.

## Acknowledgments and Disclosure of Funding

Kamélia Daudel acknowledges support of the UK Defence Science and Technology Laboratory (Dstl) and and Engineering and Physical Research Council (EPSRC) under grant EP/R013616/1. This is part of the collaboration between US DOD, UK MOD and UK EPSRC under the Multidisciplinary University Research Initiative. The authors are grateful to the reviewers for valuable remarks and suggestions on the paper.

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
