# Supplementary Material for
# Mixture weights optimisation for Alpha-Divergence Variational Inference

**Kamélia Daudel**[1,2]*, **Randal Douc**[3]
1: LTCI, Télécom Paris, Institut Polytechnique de Paris, France
2: Department of Statistics, University of Oxford, United Kingdom
3: SAMOVAR, Télécom SudParis, Institut Polytechnique de Paris, France

## A

### A.1 Equivalence between (4) and (5) with $p(y) = p(y, \mathscr{D})$

- Case $\alpha = 1$ with $f_1(u) = 1 - u + u \log(u)$ for all $u > 0$. Then,

$$
\begin{aligned}
D_1(\mu K || \mathbb{P}) &= \int_{\mathsf{Y}} f_1\left(\frac{\mu k(y)}{p(y|\mathscr{D})}\right) p(y|\mathscr{D})\nu(\mathrm{d}y) \\
&= \int_{\mathsf{Y}} \mu k(y) \log\left(\frac{\mu k(y)}{p(y|\mathscr{D})}\right) \nu(\mathrm{d}y) + 0 \\
&= \int_{\mathsf{Y}} \mu k(y) \log\left(\frac{\mu k(y)}{p(y, \mathscr{D})}\right) \nu(\mathrm{d}y) + \log p(\mathscr{D}) \\
&= \int_{\mathsf{Y}} f_1\left(\frac{\mu k(y)}{p(y, \mathscr{D})}\right) p(y, \mathscr{D})\nu(\mathrm{d}y) + 1 - p(\mathscr{D}) + \log p(\mathscr{D})
\end{aligned}
$$

Thus,

$$
\inf_{\mu \in \mathsf{M}} D_1(\mu K || \mathbb{P}) \Leftrightarrow \inf_{\mu \in \mathsf{M}} \Psi_1(\mu; p) \quad \text{with} \quad p(y) = p(y, \mathscr{D})
$$

- Case $\alpha = 0$ with $f_0(u) = u - 1 - \log(u)$ for all $u > 0$.

$$
\begin{aligned}
D_0(\mu K || \mathbb{P}) &= \int_{\mathsf{Y}} f_0\left(\frac{\mu k(y)}{p(y|\mathscr{D})}\right) p(y|\mathscr{D})\nu(\mathrm{d}y) \\
&= \int_{\mathsf{Y}} -\log\left(\frac{\mu k(y)}{p(y|\mathscr{D})}\right) p(y|\mathscr{D})\nu(\mathrm{d}y) \\
&= \int_{\mathsf{Y}} -\log\left(\frac{\mu k(y)}{p(y, \mathscr{D})}\right) p(y|\mathscr{D})\nu(\mathrm{d}y) - \log p(\mathscr{D}) \\
&= \frac{1}{p(\mathscr{D})}\left[\int_{\mathsf{Y}} f_1\left(\frac{\mu k(y)}{p(y, \mathscr{D})}\right) p(y, \mathscr{D})\nu(\mathrm{d}y) + p(\mathscr{D}) - 1 - p(\mathscr{D})\log p(\mathscr{D})\right]
\end{aligned}
$$

Thus

$$
\inf_{\mu \in \mathsf{M}} D_0(\mu K || \mathbb{P}) \Leftrightarrow \inf_{\mu \in \mathsf{M}} \Psi_0(\mu; p) \quad \text{with} \quad p(y) = p(y, \mathscr{D})
$$

---

*Corresponding author: kamelia.daudel@stats.ox.ac.uk

35th Conference on Neural Information Processing Systems (NeurIPS 2021).

- Case $\alpha \in \mathbb{R} \setminus \{1\}$ with $f_\alpha(u) = \frac{1}{\alpha(\alpha-1)}[u^\alpha - 1 - \alpha(u-1)]$ for all $u > 0$.

$$
\begin{aligned}
D_\alpha(&\mu K||\mathbb{P}) \\
&= \int_\mathsf{Y} f_\alpha\left(\frac{\mu k(y)}{p(y|\mathscr{D})}\right) p(y|\mathscr{D})\nu(\mathrm{d}y) \\
&= \int_\mathsf{Y} \frac{1}{\alpha(\alpha-1)}\left[\left(\frac{\mu k(y)}{p(y|\mathscr{D})}\right)^\alpha - 1\right] p(y|\mathscr{D})\nu(\mathrm{d}y) \\
&= p(\mathscr{D})^{\alpha-1}\int_\mathsf{Y} \frac{1}{\alpha(\alpha-1)}\left[\left(\frac{\mu k(y)}{p(y,\mathscr{D})}\right)^\alpha - 1\right] p(y,\mathscr{D})\nu(\mathrm{d}y) + \frac{p(\mathscr{D})^\alpha - 1}{\alpha(\alpha-1)} \\
&= p(\mathscr{D})^{\alpha-1}\int_\mathsf{Y} f_\alpha\left(\frac{\mu k(y)}{p(y,\mathscr{D})}\right) p(y,\mathscr{D})\nu(\mathrm{d}y) + \frac{\alpha p(\mathscr{D})^{\alpha-1} + (1-\alpha)p(\mathscr{D})^\alpha - 1}{\alpha(\alpha-1)}
\end{aligned}
$$

Thus,

$$
\inf_{\mu \in \mathsf{M}} D_\alpha(\mu K||\mathbb{P}) \Leftrightarrow \inf_{\mu \in \mathsf{M}} \Psi_\alpha(\mu; p) \quad \text{with} \quad p(y) = p(y, \mathscr{D})
$$

## A.2  [1, Theorem 1] with $\Gamma(v) = [(\alpha-1)v + 1]^{\eta/(1-\alpha)}$

**Theorem 4** ([1, Theorem 1] with $\Gamma(v) = [(\alpha-1)v+1]^{\eta/(1-\alpha)}$). *Assume that $p$ and $k$ are as in* (A1). *Let $\alpha \in \mathbb{R} \setminus \{1\}$, let $\kappa$ be such that $(\alpha-1)\kappa \geqslant 0$, let $\mu \in \mathrm{M}_1(\mathsf{T})$ and let $\eta \in (0,1]$ be such that*

$$
0 < \mu(\Gamma(b_{\mu,\alpha} + \kappa)) < \infty \tag{13}
$$

*holds and $\Psi_\alpha(\mu) < \infty$. Then, the two following assertions hold.*

  (i) *We have $\Psi_\alpha \circ \mathcal{I}_\alpha(\mu) \leqslant \Psi_\alpha(\mu)$.*

  (ii) *We have $\Psi_\alpha \circ \mathcal{I}_\alpha(\mu) = \Psi_\alpha(\mu)$ if and only if $\mu = \mathcal{I}_\alpha(\mu)$.*

## A.3  The case $\alpha < 1$ for the Power Descent algorithm

Let $\alpha \neq 1$, $\eta \in (0,1]$, $\kappa$ be such that $(\alpha-1)\kappa \geqslant 0$ and let the initial probability measure $\mu_1 \in \mathrm{M}_1(\mathsf{T})$ be such that $\Psi_\alpha(\mu_1) < \infty$. Recall that the Power Descent builds the sequence of probability measures $(\mu_n)_{n \in \mathbb{N}^\star}$

$$
\mu_{n+1} = \mathcal{I}_\alpha(\mu_n), \qquad n \in \mathbb{N}^\star,
$$

where for all $\mu \in \mathrm{M}_1(\mathsf{T})$, the one-step transition $\mu \mapsto \mathcal{I}_\alpha(\mu)$ is given by

$$
\mathcal{I}_\alpha(\mu)(\mathrm{d}\theta) = \frac{\mu(\mathrm{d}\theta) \cdot [(\alpha-1)(b_{\mu,\alpha}(\theta) + \kappa) + 1]^{\frac{\eta}{1-\alpha}}}{\mu([(\alpha-1)(b_{\mu,\alpha} + \kappa) + 1]^{\frac{\eta}{1-\alpha}})} \tag{14}
$$

and where for all $\theta \in \mathsf{T}$,

$$
b_{\mu,\alpha}(\theta) = \int_\mathsf{Y} k(\theta, y) f'_\alpha\left(\frac{\mu k(y)}{p(y)}\right) \nu(\mathrm{d}y).
$$

In particular, since for all $\alpha \neq 1$ and all $u > 0$, $f'_\alpha(u) = \frac{1}{\alpha-1}\left[u^{\alpha-1} - 1\right]$, we have that

$$
b_{\mu,\alpha}(\theta) = \frac{1}{\alpha-1}\int_\mathsf{Y} k(\theta, y)\left(\frac{\mu k(y)}{p(y)}\right)^{\alpha-1}\nu(\mathrm{d}y) - \frac{1}{\alpha-1}. \tag{15}
$$

Here, $b_{\mu,\alpha}(\theta)$ cannot fully be computed in closed-form, which is mainly due to the fact that this quantity involves $(\mu k(y))^{\alpha-1}$. Nevertheless and as underlined in [1], one way to bypass this problem is to introduce an unbiased estimate of $b_{\mu,\alpha}(\theta)$. Letting $Y \sim q_{IS}$, this can for example be done by considering the unbiased estimate of $b_{\mu,\alpha}(\theta)$ given by

$$
\hat{b}_{\mu,\alpha}(\theta) = \frac{1}{\alpha-1}\frac{k(\theta, Y)}{q_{IS}(Y)}\left(\frac{p(Y)}{\mu k(Y)}\right)^{1-\alpha} - \frac{1}{\alpha-1}
$$

Observe then that when $p(Y) = 0$, this estimator will not blow up as long as $\alpha < 1$ and $\mu k(Y) > 0$. For this reason, setting $q_{IS} = \mu k$ and $\alpha < 1$ can be numerically advantageous from an implementation point of view, especially for multimodal targets or whenever the support of $p$ does not contain the support of $\mu k$.

More generally, Bayesian tasks aim at computing integrals of the form

$$\int_Y h(y) p(y|\mathscr{D}) \nu(\mathrm{d}y) , \tag{16}$$

where $h$ is a function of interest defined on $Y$. A common way to approximate intractable integrals of the form (16) is to resort to Importance Sampling methods and in that case we are also interested in ensuring that the support of the variational approximation $q \in \mathcal{Q}$ (with $q = \mu k$ in our case) is included in the support of $p$. Seeking to solve the Variational Inference optimation problem

$$\inf_{\mu \in \mathsf{M}} D_\alpha(\mu K || \mathbb{P})$$

for $\alpha < 1$ enables this to happen, as opposed to the case $\alpha \geqslant 1$ for which the $\alpha$-divergenve exhibits the so-called *mode-seeking* property [2, 3, 4].

**Remark 2** (The function $\theta \mapsto b_{\mu,\alpha}(\theta)$ for the special case $\alpha = 1$). *Since for all $\theta \in \mathsf{T}$,*

$$b_{1,\alpha}(\theta) = \int_Y k(\theta, y) \log\left(\frac{\mu k(y)}{p(y)}\right) \nu(\mathrm{d}y)$$

$$= \int_Y k(\theta, y) \log\left(\mu k(y)\right) \nu(\mathrm{d}y) - \int_Y k(\theta, y) \log\left(p(y)\right) \nu(\mathrm{d}y)$$

*the second term of the r.h.s $\mathbb{E}_{k(\theta,\cdot)}\left[\log(p)\right]$ in the last equality might be computable in closed-form for specific models $p(y) = p(y, \mathscr{D})$, which is an aspect left for future work. As a whole, well-chosen samplers and variance reduction methods appear to be a necessity even in the case $\alpha = 1$ so that the obtained Monte Carlo estimator of $\theta \mapsto b_{\mu,\alpha}(\theta)$ do not suffer from a too large variance.*

# B

## B.1 Proof that (A2) is satisfied in Example 1

*Proof that* (A2) *is satisfied in Example 1.*

We have $k_h(\theta, y) = \frac{e^{-\|y-\theta\|^2/(2h^2)}}{(2\pi h^2)^{d/2}}$ and $p(y) = c \times \left[0.5 \frac{e^{-\|y-\theta_1^\star\|^2/2}}{(2\pi)^{d/2}} + 0.5 \frac{e^{-\|y-\theta_2^\star\|^2/2}}{(2\pi)^{d/2}}\right]$ for all $\theta \in \mathsf{T}$ and all $y \in \mathsf{Y}$. Recall that by assumption $\mathsf{T} = \mathcal{B}(0, r) \subset \mathbb{R}^d$ with $r > 0$. Then, for all $\alpha \in [0, 1)$, we are interested in proving

$$\int_Y \sup_{\theta \in \mathsf{T}} k(\theta, y) \times \sup_{\theta' \in \mathsf{T}} \left(\frac{k(\theta', y)}{p(y)}\right)^{\alpha-1} \nu(\mathrm{d}y) < \infty \tag{17}$$

and

$$\int_Y \sup_{\theta \in \mathsf{T}} \left| \log\left(\frac{k_h(\theta, y)}{p(y)}\right) \right| p(y) \nu(\mathrm{d}y) < \infty . \tag{18}$$

(i) We start by proving (17). First note that for all $\theta, \theta' \in \mathsf{T}$ and for all $y \in \mathsf{Y}$ we can write

$$\frac{k_h(\theta, y)}{k_h(\theta', y)} = e^{\frac{-\|y-\theta\|^2 + \|y-\theta'\|^2}{2h^2}} = e^{\frac{2<y,\theta-\theta'> - \|\theta\|^2 + \|\theta'\|^2}{2h^2}}$$

$$\leqslant e^{\frac{2|<y,\theta-\theta'>| + \|\theta\|^2 + \|\theta'\|^2}{2h^2}} \leqslant e^{\frac{\|y\|\|\theta-\theta'\| + r^2}{h^2}} .$$

from which we deduce that for all $\theta, \theta' \in \mathsf{T}$ and for all $y \in \mathsf{Y}$,

$$\frac{k_h(\theta, y)}{k_h(\theta', y)} \leqslant e^{\frac{\|y\|2r+r^2}{h^2}} \tag{19}$$

and that

$$\int_Y \sup_{\theta \in \mathsf{T}} k(\theta, y) \times \sup_{\theta' \in \mathsf{T}} \left( \frac{k(\theta', y)}{p(y)} \right)^{\alpha - 1} \nu(\mathrm{d}y) \leqslant \int_Y k(\theta, y) e^{\frac{\|y\|2r + r^2}{h^2}} \sup_{\theta' \in \mathsf{T}} \left( \frac{k(\theta', y)}{p(y)} \right)^{\alpha - 1} \nu(\mathrm{d}y).$$

Additionally, Jensen's inequality applied to the concave function $u \mapsto u^{1-\alpha}$ implies

$$\int_Y k(\theta, y) e^{\frac{\|y\|2r + r^2}{h^2}} \sup_{\theta' \in \mathsf{T}} \left( \frac{k(\theta', y)}{p(y)} \right)^{\alpha - 1} \nu(\mathrm{d}y) \leqslant \left( \int_Y k(\theta, y) e^{\frac{\|y\|2r + r^2}{(1-\alpha)h^2}} \sup_{\theta' \in \mathsf{T}} \frac{p(y)}{k(\theta', y)} \nu(\mathrm{d}y) \right)^{1-\alpha}$$

$$\leqslant \left( \int_Y \sup_{\theta, \theta' \in \mathsf{T}} \frac{k_h(\theta, y)}{k_h(\theta', y)} e^{\frac{\|y\|2r + r^2}{(1-\alpha)h^2}} p(y)\nu(\mathrm{d}y) \right)^{1-\alpha}$$

Now using (19), we can deduce

$$\int_Y \sup_{\theta, \theta' \in \mathsf{T}} \frac{k_h(\theta, y)}{k_h(\theta', y)} e^{\frac{\|y\|2r + r^2}{(1-\alpha)h^2}} p(y)\nu(\mathrm{d}y) \leqslant \int_Y e^{\frac{\|y\|2r + r^2}{h^2}\left(1 + \frac{1}{1-\alpha}\right)} p(y)\nu(\mathrm{d}y) < \infty ,$$

which yields the desired result.

(ii) We now prove (18). For all $y \in \mathsf{Y}$ and all $\theta \in \mathsf{T}$, we have

$$e^{- \sup_{\theta \in \mathsf{T}} \frac{\|y - \theta\|^2}{2h^2}} \leqslant (2\pi h^2)^{d/2} k_h(\theta, y) \leqslant 1$$

$$e^{- \max_{i \in \{1,2\}} \frac{\|y - \theta_i^\star\|^2}{2}} \leqslant c^{-1}(2\pi)^{d/2} p(y) \leqslant 1$$

and we can deduce for all $y \in \mathsf{Y}$ and all $\theta \in \mathsf{T}$

$$\left| \log \left( \frac{k_h(\theta, y)}{p(y)} \right) \right| \leqslant \sup_{\theta \in \mathsf{T}} \frac{\|y - \theta\|^2}{2h^2} + \max_{i \in \{1,2\}} \frac{\|y - \theta_i^\star\|^2}{2} + d|\log h| + |\log c|$$

$$\leqslant \frac{(\|y\| + r)^2}{2} \left[ \frac{1}{h^2} + 1 \right] + d|\log h| + |\log c| . \tag{20}$$

Since we have

$$\int_Y \left( \frac{(\|y\| + r)^2}{2} \left[ \frac{1}{h^2} + 1 \right] + d|\log h| + |\log c| \right) p(y)\nu(\mathrm{d}y) < \infty$$

we deduce that (18) holds.

$$\square$$

## B.2   Proof of Theorem 2

We start with some preliminary results. Let $\zeta, \zeta' \in \mathrm{M}_1(\mathsf{T})$. Recall that we say that $\zeta \mathcal{R} \zeta'$ if and only if $\zeta K = \zeta' K$ and that $\mathrm{M}_{1,\zeta}(\mathsf{T})$ denotes the set of probability measures dominated by $\zeta$.

**Lemma 3.** *Assume* (A1). *Let* $\mathsf{M}$ *be a convex subset of* $\mathrm{M}_1(\mathsf{T})$ *and let* $\zeta_1, \zeta_2 \in \mathrm{M}_1(\mathsf{T})$ *be such that*

$$\Psi_\alpha(\zeta_1) = \Psi_\alpha(\zeta_2) = \inf_{\zeta \in \mathsf{M}} \Psi_\alpha(\zeta).$$

*Then, we have* $\zeta_1 \mathcal{R} \zeta_2$.

*Proof.* For all $y \in \mathsf{Y}$, set $u_y = \zeta_1 k(y)/p(y)$ and $v_y = \zeta_2 k(y)/p(y)$. Then, for all $y \in \mathsf{Y}$ and for all $t \in (0, 1)$, $f_\alpha(tu_y + (1-t)v_y) \leqslant tf_\alpha(u_y) + (1-t)f_\alpha(v_y)$ by convexity of $f_\alpha$ and we obtain

$$\Psi_\alpha(t\zeta_1 + (1-t)\zeta_2) \leqslant t\Psi_\alpha(\zeta_1) + (1-t)\Psi_\alpha(\zeta_2) = \inf_{\zeta \in \mathsf{M}} \Psi_\alpha(\zeta) . \tag{21}$$

Furthermore, $t\zeta_1 + (1-t)\zeta_2 \in \mathsf{M}$ which implies that we have equality in (21).

Consequently, for all $t \in (0, 1)$ :

$$\int_Y \underbrace{[tf_\alpha(u_y) + (1-t)f_\alpha(v_y) - f_\alpha(tu_y + (1-t)v_y)]}_{\geqslant 0} p(y)\nu(\mathrm{d}y) = 0 .$$

Now using that $f_\alpha$ is strictly convex, we deduce that for $p$-almost all $y \in \mathsf{Y}$, $\zeta_1 k(y) = \zeta_2 k(y)$ that is $\zeta_1 \mathcal{R} \zeta_2$.  $\square$

**Lemma 4.** *Assume* (A1). *Let* $\alpha \in \mathbb{R} \setminus \{1\}$, *let* $\kappa$ *be such that* $(\alpha - 1)\kappa \geqslant 0$ *and let* $\mu^{\star} \in \mathrm{M}_1(\mathsf{T})$ *be a fixed point of* $\mathcal{I}_\alpha$. *Then,*

$$\Psi_\alpha(\mu^\star) = \inf_{\zeta \in \mathrm{M}_{1,\mu^\star}(\mathsf{T})} \Psi_\alpha(\zeta) . \tag{22}$$

*Furthermore, for all* $\zeta \in \mathrm{M}_{1,\mu^\star}(\mathsf{T})$, $\Psi_\alpha(\mu^\star) = \Psi_\alpha(\zeta)$ *implies that* $\mu^\star \mathcal{R}\zeta$.

*Proof.* Let $\zeta \in \mathrm{M}_{1,\mu^\star}(\mathsf{T})$ be such that $\Psi_\alpha(\zeta) \leqslant \Psi_\alpha(\mu^\star)$. We have that

$$\zeta\left(b_{\mu^\star,\alpha} - \mu^\star(b_{\mu^\star,\alpha})\right) \leqslant \Psi_\alpha(\zeta) - \Psi_\alpha(\mu^\star) \leqslant 0 . \tag{23}$$

Furthermore, since $\mu^\star$ is a fixed point of $\mathcal{I}_\alpha$, $\Gamma(b_{\mu^\star,\alpha} + \kappa)$, hence $|b_{\mu^\star,\alpha} + \kappa + 1/(\alpha - 1)|$ is $\mu^\star$-almost all constant. In addition, $b_{\mu^\star,\alpha} + \kappa + 1/(\alpha - 1)$ is of constant sign by assumption on $\kappa$. Since $\zeta \preceq \mu^\star$, we thus deduce that

$$\zeta\left(b_{\mu^\star,\alpha} - \mu^\star(b_{\mu^\star,\alpha})\right) = 0 .$$

Combining this result with (23) yields $\Psi_\alpha(\zeta) = \Psi_\alpha(\mu^\star)$ and we recover (22).

Finally, assume there exists $\zeta \in \mathrm{M}_{1,\mu^\star}(\mathsf{T})$ such that $\Psi_\alpha(\mu^\star) = \Psi_\alpha(\zeta)$. Then, since $\mathrm{M}_{1,\mu^\star}(\mathsf{T})$ is a convex set, we have by Lemma 3 that $\mu^\star \mathcal{R}\zeta$. □

We now move on to the proof of Theorem 2.

*Proof of Theorem 2.* For convenience, we define the notation $\Psi_{\alpha,\Theta}(\boldsymbol{\lambda}) := \Psi_\alpha\left(\mu_{\boldsymbol{\lambda},\Theta}\right)$ for all $\boldsymbol{\lambda} \in \mathcal{S}_J$. In this proof, we will use the equivalence relation $\mathcal{R}$ defined by: $\zeta \mathcal{R}\zeta'$ if and only if $\zeta K = \zeta' K$ and we write $\mathrm{M}_{1,\zeta}(\mathsf{T})$ the set of probability measures dominated by $\zeta$.

(i) *Any possible limit of convergent subsequence of* $(\boldsymbol{\lambda}_n)_{n \in \mathbb{N}^\star}$ *is a fixed point of* $\mathcal{I}_\alpha^{\mathrm{mixt}}$.

First note that by (A3), we have that $|\Psi_{\alpha,\Theta}(\boldsymbol{\lambda})| < \infty$ and that (13) is satisfied for all $\mu_{\boldsymbol{\lambda},\Theta}$ such that $\boldsymbol{\lambda} \in \mathcal{S}_J$. This means that the sequence $(\boldsymbol{\lambda}_n)_{n \in \mathbb{N}^\star}$ defined by (8) is well-defined, that the sequence $(\Psi_{\alpha,\Theta}(\boldsymbol{\lambda}_n))_{n \in \mathbb{N}^\star}$ is lower-bounded and that $\Psi_{\alpha,\Theta}(\boldsymbol{\lambda}_n)$ is finite for all $n \in \mathbb{N}^\star$. As $(\Psi_{\alpha,\Theta}(\boldsymbol{\lambda}_n))_{n \in \mathbb{N}^\star}$ is nonincreasing by Theorem 4-(i), it converges in $\mathbb{R}$ and in particular we have

$$\lim_{n \to \infty} \Psi_{\alpha,\Theta} \circ \mathcal{I}_\alpha^{\mathrm{mixt}}(\boldsymbol{\lambda}_n) - \Psi_{\alpha,\Theta}(\boldsymbol{\lambda}_n) = 0 .$$

Let $(\boldsymbol{\lambda}_{\varphi(n)})_{n \in \mathbb{N}^\star}$ be a convergent subsequence of $(\boldsymbol{\lambda}_n)_{n \in \mathbb{N}^\star}$ and denote by $\bar{\boldsymbol{\lambda}}$ its limit. Since the function $\boldsymbol{\lambda} \mapsto \Psi_{\alpha,\Theta} \circ \mathcal{I}_\alpha^{\mathrm{mixt}}(\boldsymbol{\lambda}) - \Psi_{\alpha,\Theta}(\boldsymbol{\lambda})$ is continuous we obtain that $\Psi_{\alpha,\Theta} \circ \mathcal{I}_\alpha^{\mathrm{mixt}}(\bar{\boldsymbol{\lambda}}) = \Psi_{\alpha,\Theta}(\bar{\boldsymbol{\lambda}})$ and hence by Theorem 4-(ii), $\bar{\boldsymbol{\lambda}}$ is a fixed point of $\mathcal{I}_\alpha^{\mathrm{mixt}}$.

(ii) *The set* $F = \left\{\boldsymbol{\lambda} \in \mathcal{S}_J \ : \ \boldsymbol{\lambda} = \mathcal{I}_\alpha^{\mathrm{mixt}}(\boldsymbol{\lambda})\right\}$ *of fixed points of* $\mathcal{I}_\alpha^{\mathrm{mixt}}$ *is finite.*

For any subset $R \subset \{1, \ldots, J\}$, define

$$\mathcal{S}_{J,R} = \{\boldsymbol{\lambda} \in \mathcal{S}_J \ : \ \forall i \in R^c, \lambda_i = 0, \forall j \in R^c, \lambda_j \neq 0\} ,$$
$$\tilde{\mathcal{S}}_{J,R} = \{\boldsymbol{\lambda} \in \mathcal{S}_J \ : \ \forall i \in R^c, \lambda_i = 0\} ,$$

and write

$$F = \bigcup_{R \subset \{1,\ldots,J\}} (S_{J,R} \cap F) .$$

In order to show that $F$ is finite, we prove by contradiction that for any $R \subset \{1, \ldots, J\}$, $S_{J,R} \cap F$ contains at most one element. Assume indeed the existence of two distinct elements $\boldsymbol{\lambda} \neq \boldsymbol{\lambda}'$ belonging to $S_{J,R} \cap F$. Since $\mathrm{M}_{1,\mu_{\boldsymbol{\lambda},\Theta}}(\mathsf{T}) = \mathrm{M}_{1,\mu_{\boldsymbol{\lambda}',\Theta}}(\mathsf{T}) = \left\{\mu_{\boldsymbol{\lambda}'',\Theta} \ : \ \boldsymbol{\lambda}'' \in \tilde{\mathcal{S}}_{J,R}\right\}$, Lemma 4 implies that

$$\Psi_{\alpha,\Theta}(\boldsymbol{\lambda}) = \inf_{\boldsymbol{\lambda}'' \in \tilde{\mathcal{S}}_{J,R}} \Psi_{\alpha,\Theta}\left(\boldsymbol{\lambda}''\right) = \Psi_{\alpha,\Theta}(\boldsymbol{\lambda}') .$$

Applying again Lemma 4, we get $\mu_{\boldsymbol{\lambda},\Theta} \mathcal{R} \mu_{\boldsymbol{\lambda}',\Theta}$, that is, $\mu_{\boldsymbol{\lambda},\Theta} K = \mu_{\boldsymbol{\lambda}',\Theta} K$. This means that $\sum_{j=1}^J (\lambda_j - \lambda_j') K(\theta_j, \cdot)$ is the null measure, which in turns implies the identity $\boldsymbol{\lambda} = \boldsymbol{\lambda}'$ since the family of measures $\{K(\theta_1, \cdot), \ldots, K(\theta_J, \cdot)\}$ is assumed to be linearly independent.

(iii) *Conclusion.*

According to Lemma 3 applied to the convex subset of measures $\mathsf{M} = \mathcal{S}_J$, the function $\Psi_{\alpha,\Theta}$ attains its global infimum at a unique $\boldsymbol{\lambda}_\star \in \mathcal{S}_J$. The uniqueness of $\boldsymbol{\lambda}_\star$ actually follows from the fact that, as shown above, $\mu_{\boldsymbol{\lambda},\Theta} \mathcal{R} \mu_{\boldsymbol{\lambda}',\Theta}$ if and only if $\boldsymbol{\lambda} = \boldsymbol{\lambda}'$. Then, by Theorem 4-(i) and by definition of $\boldsymbol{\lambda}_\star$

$$\Psi_{\alpha,\Theta} \circ \mathcal{I}_\alpha^{\mathrm{mixt}}(\boldsymbol{\lambda}_\star) \leqslant \Psi_{\alpha,\Theta}(\boldsymbol{\lambda}_\star) = \inf_{\boldsymbol{\lambda}' \in \mathcal{S}_J} \Psi_{\alpha,\Theta}(\boldsymbol{\lambda}') \leqslant \Psi_{\alpha,\Theta} \circ \mathcal{I}_\alpha^{\mathrm{mixt}}(\boldsymbol{\lambda}_\star)\,,$$

and hence, $\Psi_{\alpha,\Theta} \circ \mathcal{I}_\alpha^{\mathrm{mixt}}(\boldsymbol{\lambda}_\star) = \Psi_{\alpha,\Theta}(\boldsymbol{\lambda}_\star)$, showing that $\boldsymbol{\lambda}_\star \in F$ by Theorem 4-(ii). Since by (ii), $F$ is finite, there exists $L \geqslant 1$ such that $F = \left\{ \boldsymbol{\lambda}^\ell : 1 \leqslant \ell \leqslant L \right\}$, where for $i \neq j$, $\boldsymbol{\lambda}^i \neq \boldsymbol{\lambda}^j$. Without any loss of generality, we set $\boldsymbol{\lambda}^1 = \boldsymbol{\lambda}_\star$ to simplify the notation.

We now introduce a sequence $(W_\ell)_{1 \leqslant \ell \leqslant L}$ of disjoint open neighborhoods of $(\boldsymbol{\lambda}^\ell)_{1 \leqslant \ell \leqslant L}$ such that for any $\ell \in \{1, \ldots, L\}$,

$$\mathcal{I}_\alpha^{\mathrm{mixt}}(W_\ell) \cap \left( \bigcup_{j \neq \ell} W_j \right) = \emptyset \tag{24}$$

This is possible since $\mathcal{I}_\alpha^{\mathrm{mixt}}(\boldsymbol{\lambda}^\ell) = \boldsymbol{\lambda}^\ell$ and $\boldsymbol{\lambda} \mapsto \mathcal{I}_\alpha^{\mathrm{mixt}}(\boldsymbol{\lambda})$ is continuous.

By (i), the set $F$ contains all the possible limits of any subsequence of $(\boldsymbol{\lambda}_n)_{n \in \mathbb{N}^\star}$. As a consequence, there exists $N > 0$ such that for all $n \geqslant N$, $\boldsymbol{\lambda}_n \in \bigcup_{1 \leqslant \ell \leqslant L} W_\ell$. Combining with (24), there exists $\ell \in \{1, \ldots, L\}$ such that for all $n \geqslant N$, $\boldsymbol{\lambda}_n \in W_\ell$. Therefore $\boldsymbol{\lambda}^\ell$ is the only possible limit of any convergent subsequence of $(\boldsymbol{\lambda}_n)_{n \in \mathbb{N}^\star}$ and as a consequence, $\lim_{n \to \infty} \boldsymbol{\lambda}_n = \boldsymbol{\lambda}^\ell$.

Thus, the sequence $(\mu_{\boldsymbol{\lambda}_n,\Theta})_{n \in \mathbb{N}^\star}$ weakly converges to $\mu_{\boldsymbol{\lambda}^\ell,\Theta}$ as $n \to \infty$ and Theorem 1 can be applied. Since $\boldsymbol{\lambda}_1 \in \mathcal{S}_J^+$, we have $\mathrm{M}_{1,\mu_{\boldsymbol{\lambda}_1,\Theta}}(\mathsf{T}) = \left\{ \mu_{\boldsymbol{\lambda}',\Theta} : \boldsymbol{\lambda}' \in \mathcal{S}_J \right\}$ and Theorem 1-(iii) then shows that $\mu_{\boldsymbol{\lambda}^\ell,\Theta}$ is the global arginf of $\Psi_\alpha$ over all $\left\{ \mu_{\boldsymbol{\lambda}',\Theta} : \boldsymbol{\lambda}' \in \mathcal{S}_J \right\}$. Therefore, $\ell = 1$, i.e., $\boldsymbol{\lambda}^\ell = \boldsymbol{\lambda}^1 = \boldsymbol{\lambda}_\star$ and

$$\Psi_{\alpha,\Theta}(\boldsymbol{\lambda}_\star) = \inf_{\boldsymbol{\lambda}' \in \mathcal{S}_J} \Psi_{\alpha,\Theta}(\boldsymbol{\lambda}')\,.$$

$\square$

## B.3 The Power Descent for mixture models: practical version

The algorithm below provides one possible approximated version of the Power Descent algorithm. We also refer to Appendix A.3 for details regarding why the case $\alpha < 1$ is crucial when we work with approximated versions of the Power Descent algorithm.

**Algorithm 1:** *Practical version of the Power Descent for mixture models*

---

**Input:** $p$: measurable positive function, $K$: Markov transition kernel, $\alpha$: $\alpha$-divergence hyperparameter (must be different from 1), $\kappa$: hyperparameter that is such that $(\alpha - 1)\kappa \geqslant 0$, $M$: number of samples, $\Theta = \{\theta_1, \ldots, \theta_J\} \subset \mathsf{T}$: parameter set, $\Gamma(v) = [(\alpha - 1)v + 1]^{\frac{\eta}{1-\alpha}}$: function in the $(\alpha, \Gamma)$-descent, $\eta \in (0, 1]$: learning rate, $N$: total number of iterations.

**Output:** Optimised weights $\boldsymbol{\lambda}$.

Set $\boldsymbol{\lambda} = [\lambda_{1,1}, \ldots, \lambda_{J,1}]$.

**for** $n = 1 \ldots N$ **do**

> Sampling step : Draw independently $M$ samples $Y_1, \ldots, Y_M$ from $\mu_{\boldsymbol{\lambda},\Theta}k$.
>
> Expectation step : Compute $\boldsymbol{B_\lambda} = (b_j)_{1 \leqslant j \leqslant J}$ where for all $j = 1 \ldots J$
>
> $$b_j = \frac{1}{M(\alpha - 1)} \sum_{m=1}^{M} \frac{k(\theta_j, Y_m)}{\mu_{\boldsymbol{\lambda},\Theta}k(Y_m)} \left( \frac{\mu_{\boldsymbol{\lambda},\Theta}k(Y_m)}{p(Y_m)} \right)^{\alpha-1} - \frac{1}{\alpha - 1}$$
>
> and deduce $\boldsymbol{W_\lambda} = (\lambda_j \Gamma(b_j + \kappa))_{1 \leqslant j \leqslant J}$ and $w_{\boldsymbol{\lambda}} = \sum_{j=1}^{J} \lambda_j \Gamma(b_j + \kappa)$.
>
> Iteration step : Set
>
> $$\boldsymbol{\lambda} \leftarrow \frac{1}{w_{\boldsymbol{\lambda}}} \boldsymbol{W_\lambda}$$

---

# C

## C.1   Proof of Proposition 1

We first state (D1), which summarises the necessary convergence and differentiability assumptions needed in the proof of Proposition 1.

(D1) For some $\varepsilon > 0$: for all $\alpha \in [1 - \varepsilon, 1)$ or $\alpha \in (1, 1 + \varepsilon]$,

(i) there exists a function $N : \mathsf{Y} \to (0, +\infty)$ satisfying: $\int_{\mathsf{Y}} N(y)\nu(\mathrm{d}y) < \infty$ and

$$\sup_{\theta \in \mathsf{T}} k(\theta, \cdot) \times \sup_{\theta' \in \mathsf{T}} \left( \frac{k(\theta', \cdot)}{p(\cdot)} \right)^{\alpha-1} < N(\cdot) \, ;$$

(ii) there exists a function $M : \mathsf{Y} \to (0, +\infty)$ satisfying: $\int_{\mathsf{Y}} M(y)\nu(\mathrm{d}y) < \infty$ and

$$\sup_{\theta \in \mathsf{T}} k(\theta, \cdot) \times \sup_{\theta' \in \mathsf{T}} \left| \log \left( \frac{k(\theta', \cdot)}{p(\cdot)} \right) \right| \times \sup_{\theta'' \in \mathsf{T}} \left( \frac{k(\theta'', \cdot)}{p(\cdot)} \right)^{\alpha-1} < M(\cdot) \, ;$$

(iii) for all $y \in \mathsf{Y}$, we have $\int_{\mathsf{Y}} \inf_{\theta \in \mathsf{T}} k(\theta, y) \times \inf_{\theta' \in \mathsf{T}} \left( \frac{k(\theta', y)}{p(y)} \right)^{\alpha-1} \nu(\mathrm{d}y) > 0.$

Note that Assumption (D1)-(iii) is only required when $\alpha > 1$ to ensure that the quantity $[(\alpha - 1)(b_{\mu,\alpha} + \kappa) + 1]^{\frac{\eta}{1-\alpha}}$ is bounded from above. This assumption could also be replaced by the assumption that $\kappa$ is such that $(\alpha - 1)\kappa > 0$.

*Proof of Proposition 1.* For all $\theta \in \mathsf{T}$, the Dominated Convergence Theorem and (D1)-(i) yield

$$\lim_{\alpha \to 1} (\alpha - 1)(b_{\mu,\alpha}(\theta) + \kappa) + 1 = \lim_{\alpha \to 1} \int_{\mathsf{Y}} k(\theta, y) \left( \frac{\mu k(y)}{p(y)} \right)^{\alpha-1} \nu(\mathrm{d}y) + 0 = 1 \, .$$

Then, using (D1)-(ii) we have that for all $\theta \in \mathsf{T}$,

$$\lim_{\alpha \to 1} \left[ (\alpha - 1)(b_{\mu,\alpha}(\theta) + \kappa) + 1 \right]^{\frac{\eta}{1-\alpha}} = \exp \left( \lim_{\alpha \to 1} -\eta \frac{\log \left[ (\alpha - 1)(b_{\mu,\alpha}(\theta) + \kappa) + 1 \right]}{\alpha - 1} \right)$$

$$= \exp \left( \lim_{\alpha \to 1} -\eta \frac{\int_{\mathsf{Y}} k(\theta, y) \left( \frac{\mu k(y)}{p(y)} \right)^{\alpha - 1} \log \left( \frac{\mu k(y)}{p(y)} \right) \nu(\mathrm{d}y) + \kappa}{\int_{\mathsf{Y}} k(\theta, y) \left( \frac{\mu k(y)}{p(y)} \right)^{\alpha - 1} \nu(\mathrm{d}y) + (\alpha - 1)\kappa} \right)$$

$$= \exp \left[ -\eta \int_{\mathsf{Y}} k(\theta, y) \log \left( \frac{\mu k(y)}{p(y)} \right) \nu(\mathrm{d}y) \right] \exp \left( -\eta \kappa \right)$$

In addition, by the Dominated Convergence Theorem (and (D1)-(iii) when $\alpha > 1$), we have

$$\lim_{\alpha \to 1} \mu \left( \left[ (\alpha - 1)(b_{\mu,\alpha} + \kappa) + 1 \right]^{\frac{\eta}{1-\alpha}} \right) = \mu \left( \exp \left[ -\eta \int_{\mathsf{Y}} k(\cdot, y) \log \left( \frac{\mu k(y)}{p(y)} \right) \nu(\mathrm{d}y) \right] \right) \exp \left( -\eta \kappa \right) \ .$$

Thus,

$$\lim_{\alpha \to 1} [\mathcal{I}_\alpha(\mu)](h) = \int_{\mathsf{T}} \frac{\mu(\mathrm{d}\theta) h(\theta) e^{-\eta \int_{\mathsf{Y}} k(\theta, y) \log \left( \frac{\mu k(y)}{p(y)} \right) \nu(\mathrm{d}y)}}{\mu \left( e^{-\eta \int_{\mathsf{Y}} k(\cdot, y) \log \left( \frac{\mu k(y)}{p(y)} \right) \nu(\mathrm{d}y)} \right)} = [\mathcal{I}_1(\mu)](h) \ .$$

$\square$

## C.2   Derivation of the update formula for the Rényi Descent

For all $\alpha \in \mathbb{R} \setminus \{0, 1\}$ and $\kappa$ such that $(\alpha - 1)\kappa \geqslant 0$, we are interested applying the Entropic Mirror Descent algorithm to the following objective function

$$\Psi_\alpha^{AR}(\mu; p) := \frac{1}{\alpha(\alpha - 1)} \log \left( \int_{\mathsf{Y}} \mu k(y)^\alpha p(y)^{1-\alpha} \nu(\mathrm{d}y) + (\alpha - 1)\kappa \right) \ ,$$

where we will drop the dependency in $p$ in the following for convenience.

**Lemma 5.** *Assume* (A1)*. The gradient of* $\Psi_\alpha^{AR}(\mu)$ *is given by* $\theta \mapsto \frac{b_{\mu,\alpha}(\theta) + 1/(\alpha - 1)}{(\alpha - 1)(\mu(b_{\mu,\alpha}) + \kappa) + 1}$.

*Proof.* Let $\varepsilon > 0$ be small and let $\mu, \mu' \in \mathrm{M}_1(\mathsf{T})$. Then,

$$\Psi_\alpha^{AR}(\mu + \varepsilon \mu') = \frac{1}{\alpha(\alpha - 1)} \log \left( \int_{\mathsf{Y}} [(\mu + \varepsilon \mu')k(y)]^\alpha p(y)^{1-\alpha} \nu(\mathrm{d}y) + (\alpha - 1)\kappa \right)$$

$$= \frac{1}{\alpha(\alpha - 1)} \log \left( \int_{\mathsf{Y}} \mu k(y)^\alpha \left[ 1 + \alpha \varepsilon \frac{\mu' k(y)}{\mu k(y)} \right] p(y)^{1-\alpha} \nu(\mathrm{d}y) + (\alpha - 1)\kappa + o(\varepsilon) \right)$$

where we used that $(1 + u)^\alpha = 1 + \alpha u + o(u)$ as $u \to 0$. Thus,

$$\Psi_\alpha^{AR}(\mu + \varepsilon \mu') = \Psi_\alpha^{AR}(\mu) + \frac{1}{\alpha(\alpha - 1)} \log \left( 1 + \alpha \varepsilon \frac{\int_{\mathsf{Y}} \mu' k(y) \left( \frac{\mu k(y)}{p(y)} \right)^{\alpha - 1} \nu(\mathrm{d}y)}{\int_{\mathsf{Y}} \mu k(y)^\alpha p(y)^{1-\alpha} \nu(\mathrm{d}y) + (\alpha - 1)\kappa} + o(\varepsilon) \right)$$

$$= \Psi_\alpha^{AR}(\mu) + \varepsilon \frac{1}{\alpha - 1} \frac{\int_{\mathsf{Y}} \mu' k(y) \left( \frac{\mu k(y)}{p(y)} \right)^{\alpha - 1} \nu(\mathrm{d}y)}{\int_{\mathsf{Y}} \mu k(y)^\alpha p(y)^{1-\alpha} \nu(\mathrm{d}y) + (\alpha - 1)\kappa} + o(\varepsilon)$$

$$= \Psi_\alpha^{AR}(\mu) + \varepsilon \int_{\mathsf{T}} \mu'(\mathrm{d}\theta) \frac{1}{\alpha - 1} \frac{b_{\mu,\alpha}(\theta) + 1/(\alpha - 1)}{\mu(b_{\mu,\alpha}) + \kappa + 1/(\alpha - 1)} + o(\varepsilon)$$

using that $\log(1 + u) = u + o(u)$ as $u \to 0$. $\square$

Consequently, the iterative update formula for the Entropic Mirror Descent applied to the objective function $\Psi_\alpha^{AR}$ is given by

$$\mu_{n+1}(\mathrm{d}\theta) = \mu_n(\mathrm{d}\theta) \frac{e^{-\frac{\eta}{\alpha - 1} \frac{b_{\mu_n,\alpha}(\theta)}{\mu_n(b_{\mu_n,\alpha}) + \kappa + 1/(\alpha - 1)}}}{\mu_n \left( e^{-\frac{\eta}{\alpha - 1} \frac{b_{\mu_n,\alpha}}{\mu_n(b_{\mu_n,\alpha}) + \kappa + 1/(\alpha - 1)}} \right)} \ , \quad n \in \mathbb{N}^\star \ .$$

## C.3 Proof of Theorem 3

As we shall see, the proof can be adapted from the proof of [1, Theorem 2]. For all $\mu \in \mathrm{M}_1(\mathsf{T})$, we will use the notation

$$\mathcal{I}_\alpha^{AR}(\mu)(\mathrm{d}\theta) = \frac{\mu(\mathrm{d}\theta) \exp\left[-\eta \frac{b_{\mu,\alpha}(\theta)}{(\alpha-1)(\mu(b_{\mu,\alpha})+\kappa)+1}\right]}{\mu\left(\exp\left[-\eta \frac{b_{\mu,\alpha}}{(\alpha-1)(\mu_n(b_{\mu,\alpha})+\kappa)+1}\right]\right)}$$

to designate the one-step transition of the Rényi Descent algorithm. Note in passing that for all $\kappa' \in \mathbb{R}$, this definition can also be rewritten under the form

$$\mathcal{I}_\alpha^{AR}(\mu)(\mathrm{d}\theta) = \frac{\mu(\mathrm{d}\theta) \exp\left[-\eta \frac{b_{\mu,\alpha}(\theta)}{(\alpha-1)(\mu(b_{\mu,\alpha})+\kappa)+1} + \kappa'\right]}{\mu\left(\exp\left[-\eta \frac{b_{\mu,\alpha}}{(\alpha-1)(\mu_n(b_{\mu,\alpha})+\kappa)+1} + \kappa'\right]\right)} \ .$$

We also define

$$
\begin{aligned}
L &= \eta^2 \sup_{v \in \mathrm{Dom}_\alpha^{AR}} e^{-\eta v} \\
L_{\alpha,1} &= \inf_{v \in \mathrm{Dom}_\alpha^{AR}} \{1 - \eta(\alpha-1)(v-\kappa')\} \times \eta \inf_{v \in \mathrm{Dom}_\alpha^{AR}} e^{-\eta v} \\
L_{\alpha,2} &= \eta^{-1} \sup_{\theta \in \mathsf{T}, \mu \in \mathrm{M}_1(\mathsf{T})} [(\alpha-1)(b_{\mu,\alpha}(\theta)+\kappa)+1] \\
L_{\alpha,3} &= \sup_{v \in \mathrm{Dom}_\alpha^{AR}} e^{\eta v} \ .
\end{aligned}
\tag{25}
$$

### C.3.1 Recalling [1, Lemma 5]

Let $(\zeta, \mu)$ be a couple of probability measures where $\zeta$ is dominated by $\mu$ which we denote by $\zeta \preceq \mu$ and define

$$A_\alpha := \int_\mathsf{Y} \nu(\mathrm{d}y) \int_\mathsf{T} \mu(\mathrm{d}\theta) k(\theta,y) f_\alpha'\left(\frac{g(\theta)\mu k(y)}{p(y)}\right)[1-g(\theta)] \ , \tag{26}$$

where $g$ is the density of $\zeta$ w.r.t $\mu$, i.e. $\zeta(\mathrm{d}\theta) = \mu(\mathrm{d}\theta)g(\theta)$. We recall [1, Lemma 5] in Lemma 6 below.

**Lemma 6.** *[1, Lemma 5] Assume* (A1). *Then, for all* $\mu, \zeta \in \mathrm{M}_1(\mathsf{T})$ *such that* $\zeta \preceq \mu$ *and* $\Psi_\alpha(\mu) < \infty$, *we have*

$$A_\alpha \leqslant \Psi_\alpha(\mu) - \Psi_\alpha(\zeta) \ . \tag{27}$$

*Moreover, equality holds in* (27) *if and only if* $\zeta = \mu$.

### C.3.2 Adaptation of [1, Theorem 1]

**Lemma 7.** *Assume* (A1) *and* (A4). *Let* $\alpha \in \mathbb{R} \setminus \{1\}$, *let* $\kappa$ *be such that* $(\alpha-1)\kappa \geqslant 0$ *and let* $\mu \in \mathrm{M}_1(\mathsf{T})$ *be such that*

$$0 < \mu\left\{\exp\left(-\eta \frac{b_{\mu,\alpha} + 1/(\alpha-1)}{(\alpha-1)(\mu(b_{\mu,\alpha})+\kappa)+1}\right)\right\} < \infty \tag{28}$$

*holds and* $\Psi_\alpha(\mu) < \infty$. *Then, the two following assertions hold.*

  *(i) We have* $\Psi_\alpha \circ \mathcal{I}_\alpha^{AR}(\mu) \leqslant \Psi_\alpha(\mu)$.

  *(ii) We have* $\Psi_\alpha \circ \mathcal{I}_\alpha^{AR}(\mu) = \Psi_\alpha(\mu)$ *if and only if* $\mu = \mathcal{I}_\alpha^{AR}(\mu)$.

*Proof.* The proof builds on the proof of [1, Theorem 1] in the particular case $\alpha \in \mathbb{R} \setminus \{1\}$. Indeed, in this case,

$$
\begin{aligned}
A_\alpha &= \int_Y \nu(\mathrm{d}y) \int_T \mu(\mathrm{d}\theta) k(\theta, y) \frac{1}{\alpha - 1} \left[ \left( \frac{g(\theta)\mu k(y)}{p(y)} \right)^{\alpha - 1} - 1 \right] [1 - g(\theta)] \\
&= \int_Y \nu(\mathrm{d}y) \int_T \mu(\mathrm{d}\theta) k(\theta, y) \frac{1}{\alpha - 1} \left( \frac{\mu k(y)}{p(y)} \right)^{\alpha - 1} g(\theta)^{\alpha - 1} [1 - g(\theta)] \\
&= \int_T \mu(\mathrm{d}\theta) \left[ b_{\mu,\alpha}(\theta) + \frac{1}{\alpha - 1} \right] g(\theta)^{\alpha - 1} [1 - g(\theta)] \ .
\end{aligned}
$$

so that

$$
A_\alpha = [(\alpha - 1)(\mu(b_{\mu,\alpha}) + \kappa) + 1] \times \int_T \mu(\mathrm{d}\theta) \frac{b_{\mu,\alpha}(\theta) + \frac{1}{\alpha - 1}}{(\alpha - 1)(\mu(b_{\mu,\alpha}) + \kappa) + 1} g(\theta)^{\alpha - 1} [1 - g(\theta)]
$$

where $(\alpha - 1)(\mu(b_{\mu,\alpha}) + \kappa) + 1 > 0$ under (A1). Set

$$
g = \tilde{\Gamma} \circ \left( \frac{b_{\mu,\alpha} + 1/(\alpha - 1)}{(\alpha - 1)(\mu(b_{\mu,\alpha}) + \kappa) + 1} \right)
$$

where for all $v \in \mathrm{Dom}_\alpha^{AR}$,

$$
\tilde{\Gamma}(v) = \frac{e^{-\eta v}}{\mu \left\{ \exp \left( -\eta \frac{b_{\mu,\alpha} + 1/(\alpha - 1)}{(\alpha - 1)(\mu(b_{\mu,\alpha}) + \kappa) + 1} - \eta \kappa' \right) \right\}} \ .
$$

Finally, let us consider the probability space $(T, \mathcal{T}, \mu)$ and let $V$ be the random variable

$$
V(\theta) = \frac{b_{\mu,\alpha}(\theta) + 1/(\alpha - 1)}{(\alpha - 1)(\mu(b_{\mu,\alpha}) + \kappa) + 1} + \kappa' \ .
$$

Then, we have $\mathbb{E}[1 - \tilde{\Gamma}(V)] = 0$ and we can write

$$
\begin{aligned}
A_\alpha &= [(\alpha - 1)(\mu(b_{\mu,\alpha}) + \kappa) + 1] \times \mathbb{E}[(V - \kappa')\tilde{\Gamma}^{\alpha - 1}(V)(1 - \tilde{\Gamma}(V))] \\
&= [(\alpha - 1)(\mu(b_{\mu,\alpha}) + \kappa) + 1] \times \mathbb{Cov}((V - \kappa')\tilde{\Gamma}^{\alpha - 1}(V), 1 - \tilde{\Gamma}(V)) \ . \quad (29)
\end{aligned}
$$

Under (A4) with $\alpha \in \mathbb{R} \setminus \{1\}$, $v \mapsto (v - \kappa')\tilde{\Gamma}^{\alpha - 1}(v)$ and $v \mapsto 1 - \tilde{\Gamma}(v)$ are increasing on $\mathrm{Dom}_\alpha^{AR}$ which implies $\mathbb{Cov}(V\tilde{\Gamma}^{\alpha - 1}(V), 1 - \tilde{\Gamma}(V)) \geqslant 0$ and thus $A_\alpha \geqslant 0$ since $(\alpha - 1)(\mu(b_{\mu,\alpha}) + \kappa) + 1 > 0$. $\square$

### C.3.3 Adaptation of [1, Lemma 6]

Consider the probability space $(T, \mathcal{T}, \mu)$ and denote by $\mathbb{Var}_\mu$ the associated variance operator.

**Lemma 8.** *Assume* (A1) *and* (A4). *Let* $\alpha \in \mathbb{R} \setminus \{1\}$, *let* $\kappa$ *be such that* $(\alpha - 1)\kappa > 0$, *and let* $\mu \in \mathrm{M}_1(T)$ *be such that* (28) *holds and* $\Psi_\alpha(\mu) < \infty$. *Then,*

$$
\frac{(\alpha - 1)\kappa L_{\alpha,1}}{2} \mathbb{Var}_\mu \left( \frac{b_{\mu,\alpha} + 1/(\alpha - 1)}{(\alpha - 1)(\mu(b_{\mu,\alpha}) + \kappa) + 1} \right) \leqslant \Psi_\alpha(\mu) - \Psi_\alpha \circ \mathcal{I}_\alpha^{AR}(\mu) \ , \quad (30)
$$

*where*

$$
L_{\alpha,1} := \inf_{v \in \mathrm{Dom}_\alpha^{AR}} \{1 - \eta(\alpha - 1)(v - \kappa')\} \times \inf_{v \in \mathrm{Dom}_\alpha^{AR}} \eta e^{-\eta v} \ .
$$

*Proof.* The proof of Lemma 8 builds on the proof of [1, Lemma 6], which can be found in the supplementary material of [1]. Using (29) combined with the fact that under (A1), $(\alpha - 1)(\mu(b_{\mu,\alpha}) + \kappa) + 1 > (\alpha - 1)\kappa > 0$

$$
\begin{aligned}
A_\alpha &= [(\alpha - 1)(\mu(b_{\mu,\alpha}) + \kappa) + 1] \times \mathbb{Cov}((V - \kappa')\tilde{\Gamma}^{\alpha - 1}(V), 1 - \tilde{\Gamma}(V)) \\
&> (\alpha - 1)\kappa \times \mathbb{Cov}((V - \kappa')\tilde{\Gamma}^{\alpha - 1}(V), 1 - \tilde{\Gamma}(V))
\end{aligned}
$$

Furthermore,

$$\mathbb{Cov}((V - \kappa')\tilde{\Gamma}^{\alpha-1}(V), 1 - \tilde{\Gamma}(V))$$

$$= \frac{1}{2}\mathbb{E}\left[((U - \kappa')\tilde{\Gamma}^{\alpha-1}(U) - (V - \kappa')\tilde{\Gamma}^{\alpha-1}(V))(-\tilde{\Gamma}(U) + \tilde{\Gamma}(V))\right]$$

$$= \frac{1}{2}\mathbb{E}\left[\frac{(U - \kappa')\tilde{\Gamma}^{\alpha-1}(U) - (V - \kappa')\tilde{\Gamma}^{\alpha-1}(V)}{U - V}\frac{-\tilde{\Gamma}(U) + \tilde{\Gamma}(V)}{U - V}(U - V)^2\right]$$

$$\geqslant \frac{L_{\alpha,1}}{2}\mathbb{Var}_\mu\left(\frac{b_{\mu,\alpha} + 1/(\alpha - 1)}{(\alpha - 1)(\mu(b_{\mu,\alpha}) + \kappa) + 1}\right)$$

and we thus obtain (30). □

### C.3.4   Adaptation of the proof of [1, Theorem 2] to obtain Theorem 3

*Proof of Theorem 3.*  The proof of Theorem 3 builds on the proof of [1, Theorem 2], which can be found in the supplementary material of [1]. We prove the assertions successively.

(i)  The proof of (i) simply consists in verifying that we can apply Lemma 7. For all $\mu \in \mathrm{M}_1(\mathsf{T})$, (28) with $\mu = \mu_n$ holds for all $n \in \mathbb{N}^\star$ by assumption on $|B|_{\infty,\alpha}$ and since at each step $n \in \mathbb{N}^\star$, Lemma 7 combined with $\Psi_\alpha(\mu_n) < \infty$ implies that $\Psi_\alpha(\mu_{n+1}) \leqslant \Psi_\alpha(\mu_n) < \infty$, we obtain by induction that $(\Psi_\alpha(\mu_n))_{n\in\mathbb{N}^\star}$ is non-increasing.

(ii)  Let $n \in \mathbb{N}^\star$, set $\Delta_n = \Psi_\alpha(\mu_n) - \Psi_\alpha(\mu^\star)$ and for all $\theta \in \mathsf{T}$, $V_n(\theta) = \frac{b_{\mu_n,\alpha}(\theta) + \frac{1}{\alpha-1}}{(\alpha-1)(\mu_n(b_{\mu_n,\alpha}) + \kappa) + 1} + \kappa'$, such that $\mathrm{d}\mu_{n+1} \propto e^{-\eta V_n}\mathrm{d}\mu_n$.

We first show that

$$\Delta_n \leqslant L_{\alpha,2}\left[\int_\mathsf{T} \log\left(\frac{\mathrm{d}\mu_{n+1}}{\mathrm{d}\mu_n}\right)\mathrm{d}\mu^\star + \frac{L}{2}\mathbb{Var}_{\mu_n}(V_n)L_{\alpha,3}\right] . \tag{31}$$

The convexity of $f_\alpha$ implies that

$$\Delta_n \leqslant \int_\mathsf{T} b_{\mu_n,\alpha}(\mathrm{d}\mu_n - \mathrm{d}\mu^\star) \tag{32}$$

$$= \int_\mathsf{T}\left(b_{\mu_n,\alpha} + \frac{1}{\alpha-1}\right)(\mathrm{d}\mu_n - \mathrm{d}\mu^\star)$$

$$= \frac{(\alpha-1)(\mu_n(b_{\mu_n,\alpha}) + \kappa) + 1}{\eta}\int_\mathsf{T}(\mu_n(\eta V_n) - \eta V_n)\mathrm{d}\mu^\star . \tag{33}$$

Then, noting that

$$-\eta V_n = \log\mu_n\left(e^{-\eta V_n}\right) + \log\left(\frac{\mathrm{d}\mu_{n+1}}{\mathrm{d}\mu_n}\right)$$

we deduce

$$\Delta_n \leqslant L_{\alpha,2}\int_\mathsf{T}\left[\mu_n(\eta V_n) + \log\mu_n\left(e^{-\eta V_n}\right) + \log\left(\frac{\mathrm{d}\mu_{n+1}}{\mathrm{d}\mu_n}\right)\right]\mathrm{d}\mu^\star . \tag{34}$$

Since $v \mapsto e^{-\eta v}$ is $L$-smooth on $\mathrm{Dom}_\alpha^{AR}$, for all $\theta \in \mathsf{T}$ and for all $n \in \mathbb{N}^\star$ we can write

$$e^{-\eta V_n(\theta)} \leqslant e^{-\eta\mu_n(V_n)} + \eta e^{-\eta\mu_n(V_n)}(V_n(\theta) - \mu_n(V_n)) + \frac{L}{2}(V_n(\theta) - \mu_n(V_n))^2$$

which in turn implies

$$\mu_n(e^{-\eta V_n}) \leqslant e^{-\eta\mu_n(V_n)} + \frac{L}{2}\mathbb{Var}_{\mu_n}(V_n) .$$

Finally, we obtain

$$\log\mu_n(e^{-\eta V_n}) \leqslant \log e^{-\eta\mu_n(V_n)} + \log\left(1 + \frac{L}{2}\frac{\mathbb{Var}_{\mu_n}(V_n)}{e^{-\eta\mu_n(V_n)}}\right) .$$

Using that $\log(1 + u) \leqslant u$ when $u \geqslant 0$ and by definition of $L_{\alpha,3}$, we deduce

$$\log \mu_n(e^{-\eta V_n}) \leqslant -\eta \mu_n(V_n) + \frac{L}{2} \mathbb{V}\mathrm{ar}_{\mu_n}(V_n) L_{\alpha,3} \; ,$$

which combined with (34) implies (31). To conclude, we apply Lemma 8 to $g = \frac{\mathrm{d}\mu_{n+1}}{\mathrm{d}\mu_n}$ and combining with (31), we obtain

$$\Delta_n \leqslant L_{\alpha,2} \left[ \int_{\mathsf{T}} \log \left( \frac{\mathrm{d}\mu_{n+1}}{\mathrm{d}\mu_n} \right) \mathrm{d}\mu^\star + \frac{L L_{\alpha,3}}{L_{\alpha,1}(\alpha-1)\kappa} (\Delta_n - \Delta_{n+1}) \right] \; ,$$

where by assumption $L_{\alpha,1}$, $L_{\alpha,2}$ and $L_{\alpha,3} > 0$. As the r.h.s involves two telescopic sums, we deduce

$$\frac{1}{N} \sum_{n=1}^{N} \Psi_\alpha(\mu_n) - \Psi_\alpha(\mu^\star) \leqslant \frac{L_{\alpha,2}}{N} \left[ KL(\mu^\star || \mu_1) - KL(\mu^\star || \mu_{N+1}) + L \frac{L_{\alpha,3}}{L_{\alpha,1}(\alpha-1)\kappa} (\Delta_1 - \Delta_{N+1}) \right]$$

and we recover (12) using (i), that $KL(\mu^\star || \mu_{N+1}) \geqslant 0$ and that $\Delta_{N+1} \geqslant 0$.

$\square$

# D

## D.1   The Rényi Descent for mixture models: practical version

The algorithm below provides one possible approximated version of the Rényi Descent algorithm.

---

**Algorithm 2:** *Practical version of the Rényi Descent for mixture models*

---

**Input:** $p$: measurable positive function, $K$: Markov transition kernel, $\alpha$: $\alpha$-divergence
hyperparameter (must be different from 1), $\kappa$: hyperparameter that is such that $(\alpha-1)\kappa \geqslant 0$
$M$: number of samples, $\Theta = \{\theta_1, \ldots, \theta_J\} \subset \mathsf{T}$: parameter set, $\Gamma(v) = e^{-\eta v}$ with learning rate
$\eta > 0$, $N$: total number of iterations.
**Output:** Optimised weights $\boldsymbol{\lambda}$.
Set $\boldsymbol{\lambda} = [\lambda_{1,1}, \ldots, \lambda_{J,1}]$.
**for** $n = 1 \ldots N$ **do**

> Sampling step :   Draw independently $M$ samples $Y_1, \ldots, Y_M$ from $\mu_{\boldsymbol{\lambda},\Theta} k$.
>
> Expectation step :   Compute $\boldsymbol{B}_{\boldsymbol{\lambda}} = (b'_j)_{1 \leqslant j \leqslant J}$ where for all $j = 1 \ldots J$
>
> $$b_j = \frac{1}{M(\alpha-1)} \sum_{m=1}^{M} \frac{k(\theta_j, Y_m)}{\mu_{\boldsymbol{\lambda},\Theta} k(Y_m)} \left( \frac{\mu_{\boldsymbol{\lambda},\Theta} k(Y_m)}{p(Y_m)} \right)^{\alpha-1} - \frac{1}{\alpha-1}$$
>
> and for all $j = 1 \ldots J$
> $$b'_j = \frac{b_j}{(\alpha-1)(\sum_{\ell=1}^{J} b_\ell + \kappa) + 1}$$
>
> and deduce $\boldsymbol{W}_{\boldsymbol{\lambda}} = (\lambda_j \Gamma(b'_j + \kappa'))_{1 \leqslant j \leqslant J}$ and $w_{\boldsymbol{\lambda}} = \sum_{j=1}^{J} \lambda_j \Gamma(b'_j + \kappa')$.
>
> Iteration step :   Set
>
> $$\boldsymbol{\lambda} \leftarrow \frac{1}{w_{\boldsymbol{\lambda}}} \boldsymbol{W}_{\boldsymbol{\lambda}}$$

---

## D.2    Plots in dimension $d < 16$

We present here plots comparing the Power Descent, the Rényi Descent and the Entropic Mirror Descent applied to $\Psi_\alpha$ in a low-dimensional setting (i.e. $d < 16$) and using the same Exploration step as in Figure 1.

Figure 3: Plotted is the average Variational Rényi bound for the Power Descent (PD), the Rényi Descent (RD) and the Entropic Mirror Descent applied to $\Psi_\alpha$ (EMD) in dimension $d = \{4, 6, 8, 10\}$ computed over 50 replicates with $\eta_0 = 0.3$ and $\alpha = 0.5$ and $M \in \{100, 200\}$.

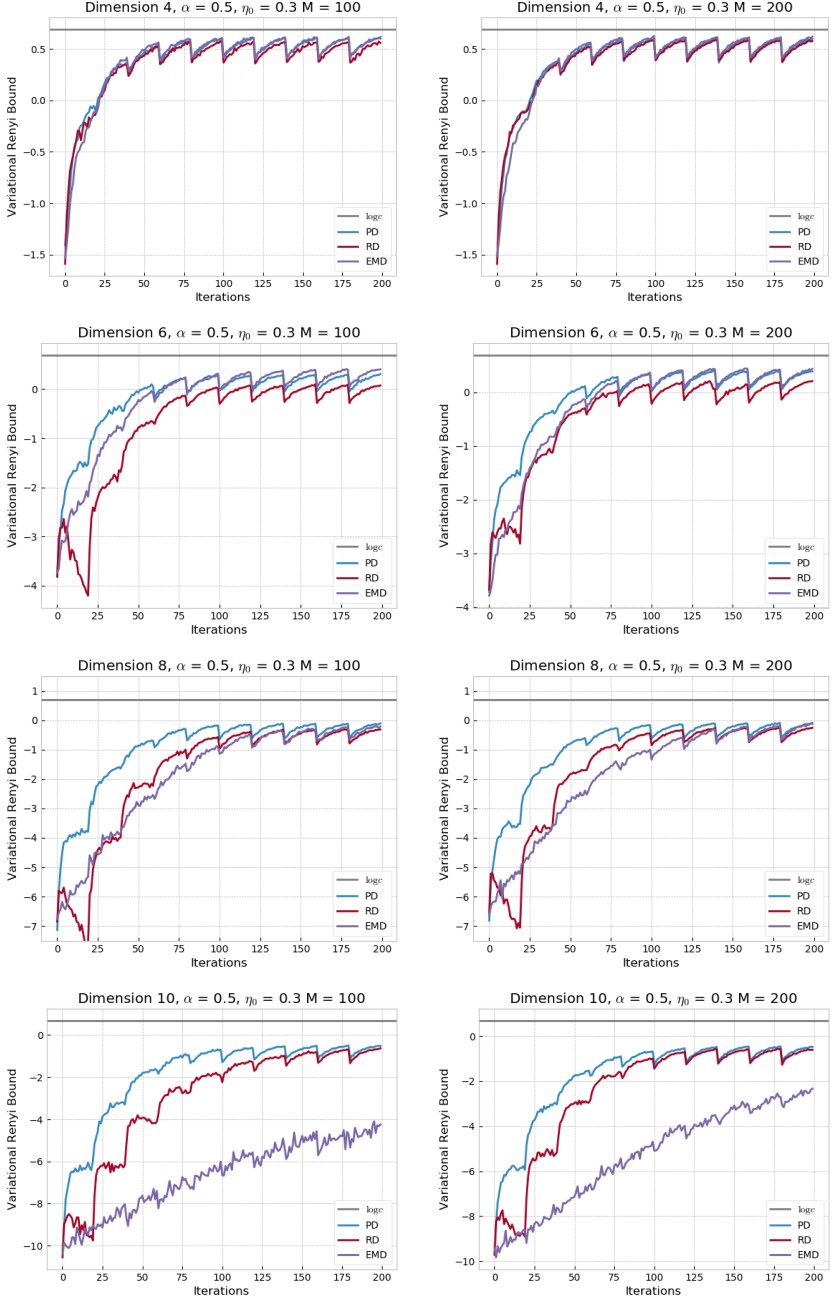

- In dimension $d = 4$, the performances are similar for the Entropic Mirror Descent applied to $\Psi_\alpha$, the Power Descent and the Rényi Descent.

- In dimension $d = 6$, the Entropic Mirror Descent applied to $\Psi_\alpha$ outperforms the Rényi Descent but is slower than the Power Descent.

- In dimension $d = 8, 10$, the Entropic Mirror Descent applied to $\Psi_\alpha$ is still able to learn, but at a much slower rate compared to the Rényi Descent and the Power Descent.

These plots notably show that the Entropic Mirror Descent applied to $\Psi_\alpha$, which is very well-supported algorithm theoretically, does work in practice in small dimensions and might even outperform the Rényi Descent (e.g. when $d = 6$).

### D.3  Alternative Exploration steps in Algorithm 2

We present here two possible alternative choices of Exploration steps in Algorithm 2, beyond the first one we have made in Section 5 and that is based on [1]. Our goal here is not to discriminate between all of them, but to illustrate the generality of the approach.

#### D.3.1  Gradient Descent

One could use a Gradient Descent approach to optimise the mixture components parameters $\{\theta_{1,t+1}, \ldots, \theta_{J,t+1}\}$ in the spirit of Rényi's $\alpha$-divergence gradient-based methods (e.g [5, 6]) or $\alpha$-divergence gradient-based methods (e.g [7, 8]).

#### D.3.2  The particular case $\alpha \in [0, 1)$

For the specific case $\alpha \in [0, 1)$ and following [9], another possibility would be to set at time $t \leqslant T$: for all $j = 1 \ldots J$

$$\theta_{j,t+1} = \operatorname{argmax}_{\theta_j \in \mathsf{T}} \int_{\mathsf{Y}} \gamma^t_{j,\alpha}(y) \log(k(\theta_j, y))\nu(\mathrm{d}y) \tag{35}$$

where for all $y \in \mathsf{Y}$,

$$\gamma^t_{j,\alpha}(y) = k(\theta_{j,t}, y) \left( \frac{\mu_{\boldsymbol{\lambda},\Theta}k(y)}{p(y)} \right)^{\alpha-1} .$$

Indeed, [9] showed that the above update formulas for $\{\theta_{1,t+1}, \ldots, \theta_{J,t+1}\}$ ensure a systematic decrease in the $\alpha$-divergence and they notably explained how these update formulas could even outperform typical Rényi's $\alpha$ / $\alpha$-divergence gradient-based approaches (we refer to [9] for details).

Furthermore, in the particular case of $d$-dimensional Gaussian density kernels with $k(\theta_{j,t}, y) = \mathcal{N}(y; m_{j,t}, \Sigma_{j,t})$ and where $\theta_{j,t} = (m_{j,t}, \Sigma_{j,t}) \in \mathsf{T}$ denotes the mean and covariance matrix of the $j$-th Gaussian component density, they obtained that the maximisation procedure (35) amounts to setting

$$\forall j = 1 \ldots J, \quad m_{j,t+1} = \frac{\int_{\mathsf{Y}} \gamma^t_{j,\alpha}(y) y \, \nu(\mathrm{d}y)}{\int_{\mathsf{Y}} \gamma^t_{j,\alpha}(y)\nu(\mathrm{d}y)}$$

$$\Sigma_{j,t+1} = \frac{\int_{\mathsf{Y}} \gamma^t_{j,\alpha}(y)(y - m_{j,t+1})(y - m_{j,t+1})^T \nu(\mathrm{d}y)}{\int_{\mathsf{Y}} \gamma^t_{j,\alpha}(y)\nu(\mathrm{d}y)} .$$

These update formulas can then always be made feasible by resorting to Monte Carlo approximations and can be used as a valid Exploration step. If we were to focus on solely updating the means $(m_{j,t+1})_{1 \leqslant j \leqslant J}$, we could for example consider the Exploration step given by:

$$\forall j = 1 \ldots J, \quad \theta_{j,t+1} = m_{j,t+1} = \frac{\sum_{m=1}^M \hat{\gamma}^{(t)}_j(Y'_m; \boldsymbol{\lambda}) \cdot Y'_m}{\sum_{m=1}^M \hat{\gamma}^{(t)}_j(Y'_m; \boldsymbol{\lambda})}$$

where the $M$ samples $(Y'_m)_{1 \leqslant m \leqslant M}$ have been drawn independently from the proposal $\mu_{\boldsymbol{\lambda},\Theta}k$ and where we have set

$$\hat{\gamma}^{(t)}_j(y; \boldsymbol{\lambda}) = \frac{k(\theta_{j,t}, y)}{\mu_{\boldsymbol{\lambda},\Theta}k(y)} \left( \frac{\mu_{\boldsymbol{\lambda},\Theta}k(y)}{p(y)} \right)^{\alpha-1} .$$

Figure 4: Plotted is the average Variational Rényi bound for the Power Descent (PD) and the Rényi Descent (RD) in dimension $d = 16$ computed over 100 replicates with $\eta_0 = 0.3$ and $\alpha = 0.5$ and an increasing number of samples $M$.

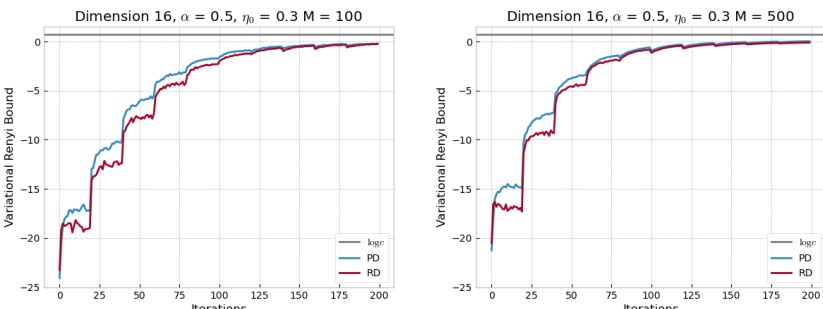

We ran Algorithm 2over 100 replicates for this choice of Exploration step with $M \in \{100, 500\}$ (and keeping the same target $p$, initial sampler $q_0$, and hyperparameters $N = 20$, $T = 10$, $\eta = \eta_0/\sqrt{N}$ with $\eta_0 = 0.3$, $\alpha = 0.5$, $J = 100$, $\kappa = 0$ and $d = 16$ as those chosen in Section 5). The results when using the Power and the Rényi Descent as Exploitation steps can be visualised in the figure below.

We then observe a similar behavior for the Power and the Rényi Descent, which illustrates the closeness between both algorithms, irrespective of the choice of the Exploration step.