# OpenReview forum: "Mixture weights optimisation for Alpha-Divergence Variational Inference"
_NeurIPS.cc/2021/Conference — NeurIPS 2021 Poster_

### Official Review · Reviewer_qaiR · 2021-07-12

**Rating:** 7
**Confidence:** 3

**Summary:**

The authors explore the connections between power descent and entropic mirror descent for variational inference with $\alpha$-divergences. The authors show that power descent converges under weaker assumptions than previously known when the weights of a mixture model are optimized. They also show a convergence rate of $O(1/N)$ for a variant of power descent, which they refer to as Rényi descent with $\alpha < 1$. A simulation experiment is included indicating the efficacy of the proposed method for reasonably high-dimensional problems, and supporting the theoretical claims.

**Limitations And Societal Impact:**

The authors address several limitations of the work, notably that practical implementations of the algorithm rely on a stochastic approximation, which is not handled in the theoretical analysis, and that convergence guarantees for power descent are not obtained in the general case. The authors do not discuss societal impacts, which seems reasonable as the contribution theoretical and not attached to a particular application.

**Main Review:**

## Summary of Review
The results presented in the paper on convergence of power descent for mixture weights, as well as convergence of Rényi-Descent appear to be interesting and original. However, as I am not familiar with some of the related work, I am hesitant to make an overly strong statement in this regard. Convergence of variational objective functions is certainly of interest to the probabilistic machine learning community. The paper was reasonably clear, though I think it could have been more self-contained. In particular,
- At points notation was used without being introduced e.g.:
    - $\mu f= \int f d\mu$
    -  $\mu K$ is the measure with $\nu$-density $\mu k$
- The paper would benefit from a, perhaps higher-level, discussion of related work on optimization of variational objectives, discussing existing work in this area.

As I am not familiar with the some of the related work, and it is possible I have misjudged some portion of the paper, I may adjust my score in light of additional information/clarification.

## General comments and questions:
- Can the authors clarify what aspects of the discussion and analysis comparing Power descent and entropic mirror descent are new to this work, and what aspects are previously contained in [17]? Is table 1 in [17] not already claiming something very much like proposition 1?
- Relatedly, can you clarify why [17, Theorem 3] does not already imply a rate of convergence rate of $O(1/N)$ in the case $\alpha=1$?
- Are there any interesting cases where $b_{\mu_n,  \alpha}$ can be computed in closed-form (so that the algorithm for which you prove guarantees can actually be implemented)?

## Typos and Minor comments:
- The main paper contains links to the appendix, that are not stated as links to the appendix. As the main paper will be separate (so these links will not work), consider stating they are in the appendix/supplement.
- Renyi $\to$ Rényi
- I think you may need an absolute continuity assumption in the definition of alpha divergence.
- Line 420: $\zeta_1 R \zeta_2$


**Time Spent Reviewing:**

5 hours

---

> ### Author Response · Authors · 2021-08-09
> **Thank you for your thorough reading.**
>
> We have corrected the typos you spotted and introduced the notation that were missing. We will add a discussion of related work on optimisation of variational objectives e.g. involving
>
> Provable Smoothness Guarantees for Black-Box Variational Inference by Justin Domke (ICML 2020), Dynamics of Coordinate Ascent Variational Inference: A Case Study in 2D Ising Models by Sean Plummer, Debdeep Pati and Anirban Bhattacharya (Entropy 2020), Consistency of variational Bayes inference for estimation and model
> selection in mixtures by Chérief-Abdellatif, B.E.; Alquier, P. (Electron. J. Stat. 2018)
>
> and are interested in any complementary references you might have in mind regarding that aspect. Please find below an answer to the questions you have asked.
>
> - *clarify what aspects of the discussion and analysis comparing Power descent and entropic mirror descent are new to this work*
>
> The extension of the Power Descent to the case $\alpha = 1$ in Proposition 1 is novel. The $O(1/N)$ convergence rate of this extension is already known from [17, Theorem 3] since the extension recovers an Entropic Mirror Descent applied to $\Psi_0$. We will clarify this in the manuscript, by notably replacing the mention {extension to $\alpha = 1$ with $O(1/N)$ convergence rate} by {extension to $\alpha = 1$} in Table 1.
>
>   Besides the extension to the case $\alpha = 1$, the two other theoretical novelties compared to [17] are (1) the full proof of the convergence for the mixture weights when $\alpha < 1$ for the Power Descent in Theorem 1 and (2) the new variant of the Power Descent that we called the Rényi Descent, for which we prove an $O(1/N)$ convergence rate in Theorem 3. Consequently, using the same numerical setting as in [17], we have added in Figure 1 new curves corresponding to the Rényi Descent. We also ran supplementary numerical experiments where we changed the Exploration step in Figure 2 of the appendix to further illustrate our claim that these two algorithms share a close connection.
>
> - *Are there any interesting cases where $b_{\mu_n, \alpha}$ can be computed in closed-form*
>
> You are making a great point. We have not found cases where $b_{\mu_n, \alpha}(\theta)$ can fully be computed in closed-form, which is mainly due to the fact that $b_{\mu_n, \alpha}(\theta)$ involves $\mu_n k$. Nevertheless, in the particular case $\alpha = 1$, since
>     $$
>     b_{\mu_n, 1}(\theta) = \int k(\theta, y) \log \left(\frac{\mu_n k(y)}{p(y)} \right) d y = \int k(\theta, y) \log \left(\mu_n k(y)\right) d y - \int k(\theta, y) \log \left({p(y)} \right) d y
>     $$
>     the second term of the r.h.s $E_{k(\theta, \cdot)}[\log(p)]$ might be computable in closed-form for specific models $p = p(\cdot, \mathcal{D})$. As a whole, our hope is to rely on well-chosen samplers and variance reduction methods so that the obtained Monte Carlo estimators of $b_{\mu_n, \alpha}(\theta)$ do not suffer from a too large variance.

---

> > ### Comment · Reviewer_qaiR · 2021-08-20
> > **Reply to authors**
> >
> > Thank you for your response. I think you addressed most of the issues that I had when reading the paper and brought up in my review. I think the suggested additions to the related work (and hopefully a small amount of high-level discussion motivating the problem!)  would improve the paper. I would encourage the authors to consider any other ways they can improve how self-contained the work is within the constraints of the relatively short conference format. I am raising my initial rating from a 6 to a 7 in light of the author response.

---

> > > ### Author Response · Authors · 2021-08-20
> > > **Thank you for your encouraging response.**
> > >
> > > We are grateful for your positive feedback and will actively work towards improving our paper as per your guidance.

---

### Official Review · Reviewer_B46Q · 2021-07-13

**Rating:** 6
**Confidence:** 3

**Summary:**

The authors study the convergence properties of the Power Descent algorithm, which has been recently proposed in [17].
Power Descent (PD) allows one to optimise the weights of a mixture model by means of $\alpha$-divergence minimisation.
In previous [17] work, PD was shown to have convergence rate O(1/N) for $\alpha>1$.
In the current work, the authors prove convergence for the case where $\alpha<1$, under less strict assumptions that in [17].
Although no convergence rate is given for $\alpha<1$, the authors propose Renyi Descent (RD) as an alternative, which is shown to have convergence rate O(1/N).
Also, PD is extended to the case $\alpha=1$, recovering thus Entropic Mirror Descent.
The convergence rate for $\alpha=1$ is shown to be O(1/N).
The convergence properties are demonstrated on a case study that involves approximating a two-component Gaussian mixture model as the target density.


**Limitations And Societal Impact:**

The authors claim that their contribution is mostly theoretical, thus it is unlikely to result in any negative societal impact.
I think this is a fair position.


**Main Review:**


The paper extends the theoretical results of [17] for Power Descent, providing a more complete characterisation of the convergence properties of $\alpha$-divergence methods.
The mixture model framework presented has been already discussed in [17]; in that sense, the contribution of the current paper is somewhat incremental.
Nevertheless, the addition of Renyi Descent as an $\alpha$-divergence optimisation method, as well as its convergence analysis could be of interest to the community.

The experimental evaluation of the paper is weak.
I appreciate that this is a mostly theoretical work, but the empirical examples are limited.
The simulation study of Section 5 could easily be extended to demonstrate the convergence properties of the methods proposed for models of different dimensionality.
For example, it is shown that Entropic Mirror Descent fails to converge for 16 dimensions, but what about a smaller dimensionality? How would it compare with Power descent or Renyi descent?

Another possibly minor issue is that certain parts of Section 2 bear many similarities to Sections 2.2, 3.1 and 3.2 in [17].
I understand that this is a background section and it has to summarise previous work, but parts of it should be rephrased.



**Time Spent Reviewing:**

12

---

> ### Author Response · Authors · 2021-08-09
> **Thank you for all your time and effort spent on reviewing our work.**
>
> We will adjust section 2 according to your remark.
>
> - *For example, it is shown that Entropic Mirror Descent fails to converge for 16 dimensions, but what about a smaller dimensionality? How would it compare with Power descent or Renyi descent?*
>
>   We observed that for the example considered in our numerical experiments (and when we average the performances over 50 trials) :
> \begin{itemize}
>    - In dimension $d = 2, 4$, the performances are similar for the Entropic Mirror Descent applied to $\Psi_\alpha$, the Power Descent and the Rényi Descent.
>    - In dimension $d = 6$, the Entropic Mirror Descent applied to $\Psi_\alpha$ outperforms the Rényi Descent but underperforms compared to the Power Descent.
>    - In dimension $d = 8, 10$, the Entropic Mirror Descent applied to $\Psi_\alpha$ is still able to learn, but at a much slower rate compared to the Rényi Descent and the Power Descent.
>
>    Thank you again for your question. If possible, we will add these plots directly in the manuscript, otherwise we will add them in the appendix.
>
> NB : we believe that your remark regarding our numerical experiments is in line with the comments from Reviewer ssFh, who suggested on the other hand that we look into higher dimensions. Consequently, we ran some additional numerical experiments with the Exploration step used to construct Figure 2 of the appendix : we observed that the Rényi Descent and the Power Descent do not break down as we increase the dimension up to $d = 100$. We plan on using these new plots as well to improve our numerical experiments section.

---

> > ### Comment · Reviewer_B46Q · 2021-08-12
> > **Interesting behaviour for Entropic Mirror Descent**
> >
> > Thank you for your response.
> > This is an interesting behaviour for Entropic Mirror Descent, but the current state of the experiments presents a rather skewed message.
> > I am now convinced that a more extensive exploration would improve this submission.

---

> > > ### Author Response · Authors · 2021-08-13
> > > **Thank you for your positive comment**
> > >
> > > We had focused on the case $d=16$ as we knew from [17] that the Entropic Mirror Descent applied to $\Psi_\alpha$ breaks down for this dimension. As our contribution was centered around providing a theoretically-sound framework for mixture weights optimisation, the case $d = 16$ permitted us to illustrate that the Rényi Descent is a more scalable alternative, with a known convergence rate, and that is closer to the Power Descent in its behavior.
> > >
> > > *Following your feedback and the feedback from Reviewer ssFh :*
> > >
> > > The plots for smaller dimensions $d < 16$ are ready to be integrated into our manuscript; that way we can insist on the fact that the Entropic Mirror Descent is a reasonable algorithm to use in small dimensions. Similarly, the plots for higher dimensions $d > 16$ up to $d = 100$ are ready to be added to our manuscript; those allow us to show that the behavior we observed in dimension $d = 16$ persists up to dimension $d = 100$ and further supports the conclusions we drew in our work.

---

### Official Review · Reviewer_ssFh · 2021-07-16

**Rating:** 6
**Confidence:** 4

**Summary:**

The authors consider the problem of variational inference with Renyi divergence for learning mixture models. They improve the analysis of the iterative method Daudel et al. Specifically, they establish fast $O(1/n)$ convergence of the method when $\alpha < 1$. They then analyze the method when $\alpha = 1$ by taking the limit of the algorithm updates. They call this algorithm Renyi descent and establish that it also converges at an $O(1/n)$ rate. They experimentally compare their approach with entropic mirror descent.

**Limitations And Societal Impact:**

No issues here.

**Main Review:**

Overall I found the content of the paper interesting. The authors complete the analysis of Daudel et al. and show fast convergence of the variety of parameter settings. The paper is theoretically interesting.

However I believe the clarity of the paper should be improved. The paper is quite dense and more care can be used to introduce concepts:

(1) The introduction in particular is quite dense. The appendix describes "practical" versions of the approach. I think a more practical introduction can improve the clarity.

(2) The paper discusses mixture models but works with a density kernel. The connection can be more clearly spelled out.

(3) The paper defines Renyi descent and discusses its connection with entropic mirror descent, however the explanation of this connection can be improved.

The experimental section is perhaps sufficient for this type of paper but not particularly interesting. The scale of the problem is quite small, only 16 dimensions. Does the method break down in higher dimensions? The authors could argue the strength of the method in general, rather than just how it compares to entropic mirror descent.

Overall, I lean towards acceptance due to the content of the paper, however I believe the clarity can be improved.

**Time Spent Reviewing:**

3

---

> ### Author Response · Authors · 2021-08-09
> **Thank you for your insightful feedback.**
>
> Your feeback has motivated us to improve the clarity of our work regarding the introduction and the connection between Rényi Descent and Entropic Mirror Descent. We agree that the link between mixture models and density kernels could be more clearly spelled out and we will insist more on this aspect in the background section to ease the reading.
>
> - *The scale of the problem is quite small, only 16 dimensions. Does the method break down in higher dimensions?*
>
> We found the case $d = 16$ interesting as we knew from [17] that the Entropic Mirror Descent applied to $\Psi_\alpha$ breaks down for the numerical examples considered in [17] and we wanted to present the Rényi Descent as an alternative that is closer to the Power Descent in its behavior.
>
> Nevertheless, you make a fair point by saying we should look into higher dimensions. Since deriving a fully adaptive algorithm entails pairing up our mixture weights update formula with an Exploration step, the choice of this Exploration step becomes increasingly important as the dimension increases.
>
> While we are not trying to discriminate between various choices of Exploration step in our manuscript (which is why the Exploration step was kept simple in the numerical experiments from Figure 1), we have outlined in the appendix some potentially suitable Exploration steps that optimise the mixture components parameters $\theta_{1,t}, \ldots, \theta_{J,t}$.
>
> Following your question, we ran our numerical experiments with the Exploration step used to construct Figure 2 of the appendix for various choices of dimension $d$. We have observed that the Rényi Descent and the Power Descent do not break down as we increase the dimension up to $d = 100$, which reinforces our conclusions. We thank you yet again for your question and, if allowed to, we will add these plots directly in the manuscript to improve on our experimental section.

---

### Official Review · Reviewer_M13e · 2021-07-18

**Rating:** 4
**Confidence:** 3

**Summary:**

This paper extends a previous work [17] in providing a convergence rate for the case of \alpha<1.

**Limitations And Societal Impact:**

Yes

**Main Review:**

My main concern is that this work is too close to the previous work [17]. The case of \alpha<1 has been discussed in the experiment of [17] in a similar setting. The proof technique is also similar. More importantly, it is not clear why the case of \alpha<1 is so crucial to the field. The exposition and literary quality in [17] is so much better than the current manuscript. I find it hard to justify a separate publication. I would suggest the authors to simply incorporate this result into [17] for publication.

**Time Spent Reviewing:**

4

---

> ### Author Response · Authors · 2021-08-09
> **Thank you for your valuable comments.  Please find below an answer to the issues you have mentioned.**
>
> - *My main concern is that this work is too close to the previous work [17]... The proof technique is also similar.*
>
> In our work, we prove three different results :
>
> 1) We first state in Theorem 2 a full convergence result for the Power Descent in the particular case of mixture weights optimisation when $\alpha <1$. This novel result completes Theorem 1 [17, Theorem 4] which assumed the existence of the limit and did not provide conditions that satisfy this assumption.
>
> 2) We obtain in Proposition 1 that the Power Descent can be extended to the case $\alpha = 1$ and that we recover an Entropic Mirror Descent algorithm. This brings us to investigate the links between Power Descent and Entropic Mirror Descent beyond the framework of [17] (in the hope of finding an algorithm that is close to the Power Descent and for which we can prove a convergence rate when $\alpha <1$).
>
> 3) We find that a relevant choice of update formula that has not been considered in [17] is given by the Rényi Descent
> $$
> \mu_{n+1}(d \theta) \propto \mu_n(d \theta) e^{- \eta \frac{b_{\mu_n, \alpha}(\theta)}{(\alpha-1)(\mu_n(b_{\mu_n,\alpha})+\kappa)+1}}, \quad n \geq 1 ~.
> $$
> Here, the name originates from the connection we establish between the Rényi Descent and Entropic Mirror Descent steps applied to the Variational Rényi bound : it allows us to classify this approach as a Rényi’s $\alpha$-divergence gradient-based approach. This is in contrast with the framework of [17], that only introduced Entropic Mirror Descent steps applied to $\Psi_\alpha$
> $$
> \mu_{n+1}(d \theta) \propto \mu_n(d \theta) e^{- \eta b_{\mu_n, \alpha}(\theta)}, \quad n \geq 1
> $$
> and that is an $\alpha$-divergence gradient-based approach. We have thus moved from a constant learning rate $\eta$ to an adaptive learning rate $\eta' = \eta [(\alpha-1)(\mu_n(b_{\mu_n,\alpha})+\kappa)+1]^{-1}$. As a consequence, it was not obvious from the framework of [17] that we would maintain $O(1/N)$ convergence rates for such a choice of adaptive learning rate. As indicated in our manuscript, we were able to obtain $O(1/N)$ convergence rates for the Rényi Descent in Theorem 3 by carefully adapting the proof of [17] to this new algorithm.
>
> - *The case of $\alpha<1$ has been discussed in the experiment of [17] in a similar setting.*
>
> We have selected on purpose a numerical framework where the Entropic Mirror Descent applied to $\Psi_\alpha$ is known to fail. Adding this time the plots for the Rényi Descent enabled us to illustrates the newly-found link between Power Descent and Rényi Descent. Additional numerical experiments where we changed the Exploration step are also available in Figure 2 of the appendix and provide further empirical justifications of our claim that the Power Descent and the Rényi Descent are connected one to another.
>
> - *it is not clear why the case of $\alpha<1$ is so crucial to the field*
>
> Thank you for pointing this out, we can clarify this in the manuscript. The case $\alpha <1$ is crucial to the field since Variational Inference methods often resort to Monte Carlo approximations in practice. In our case, letting $Y \sim q_n$, an unbiased estimate of $b_{\mu_n, \alpha}(\theta)$ is given by
> $$
> \hat{b}_{\mu_n, \alpha}(\theta) = \frac{1}{\alpha-1}\frac{k(\theta, Y)}{q_n(Y)}  \left(\frac{p(Y)}{\mu_n k(Y)} \right)^{1-\alpha} - \frac{1}{\alpha-1}
> $$
> When $p(Y) = 0$, this estimator will not blow up as long as $\alpha <1$ and $\mu_n k(Y) >0$. For this reason, setting $q_n = \mu_n k$ can be numerically advantageous, especially for multimodal targets or at the start of the algorithm if the support of $\mu_n k$ is very different from the support of $p$. In a typical Monte Carlo sampling fashion, we are also interested in ensuring that the support of $p$ is included in the support of $q_n =  \mu_n k$ and the case $\alpha < 1$ enables this to happen.

---

> > ### Comment · Reviewer_M13e · 2021-09-01
> > **Re**
> >
> > I would like to thank the authors for their reply. I went through the paper again, together with [17]. I still feel that it makes most sense to combine them as one journal paper for better exposition as well as completeness. I appreciate that the authors clarify their motivation of adopting \alpha < 1 in the Renyi divergence. The original question I had in mind was: since Renyi divergence with larger \alpha is stronger, why wouldn't one want to focus on larger \alpha. But implementation issue is a good point. For that reason, I have increased my evaluation.

---

### Decision · Program_Chairs · 2021-09-27

**Decision:**

Accept (Poster)

**Comment:**

Variational approximations are approximations of the posterior in Bayesian statistics, obtained by minimizing a divergence between the approximation and the posterior, where the approximation is constrained to belong to a given set (for example, the set of Gaussian distributions, or the set of mixtures of Gaussian distributions). The divergence chosen is usually the Kullback-Leibler divergence (KL). However, some recent works highlighted the possible benefits of using alpha-divergences instead (note that KL is recovered as the special case alpha=1). [17] proposed an iterative scheme, referred to as the Power Descent algorithm, to minimize the alpha-divergence in practice. A nice property of this scheme is that it leads to a sequence of approximations such that the divergence with respect to the posterior is decreasing.

This paper extends the analysis of the Power Descent algorithm initiated in [17] to approximations by mixtures models. The author(s) then study the convergence of the sequence of approximations to the minimizer of the divergence. In the case alpha>1, this is a direct consequence of Theorem 1 (stated in the paper, but was already proven in [17]). The convergence in the case alpha<1 is given by Theorem 2 (this is a new, and difficult result). Finally, they analyze the case alpha=1 by taking the limit of the algorithm updates. They refer to this algorithm as "Renyi descent", for which they prove a convergence in 1/N (N being the number of iterations, this is Theorem 3). They compare their approach with Entropic Mirror Descent in a set of numerical experiments.

Three Reviewers think that while this is strongly based on [17], there are useful and nontrivial extensions, and I agree with them. One of the members of the Committee already reviewed an earlier version of this paper, and was satisfied to see many of his/her comments taken into account in this version of the paper. One of them points some weaknesses in the empirical evaluation, but it clear to me that the main contribution here is theoretical. I will therefore recommend to accept the paper.

It would be nice if you could improve the writing of the paper before sending the camera ready version. Reviewer ssFh wrote the paper is too dense, Reviewer M13e found some undefined notations and Reviewer B46Q asked you to rephrase parts of Section 2, that explain the context of [17] but that are too close to the original.